# Assessing Sustainability of the Capital and Emerging Secondary Cities of Cambodia Based on the 2018 Commune Database

**Puthearath Chan** [1,2] 

1   General Secretariat of the National Council for Sustainable Development, Phnom Penh 12301, Cambodia; cptr5@hanyang.ac.kr
2   Office of Sustainable Lifestyle, Ministry of Environment, Phnom Penh 12301, Cambodia

**Abstract:** The world is rapidly urbanizing which 68% of its population is expected to live in urban areas by 2050. Likewise, secondary cities of Cambodia are rapidly emerging while the capital is the largest city with a population of more than two million. Improving urban sustainability is, therefore, necessary for the world, as well as Cambodia. Thus, Cambodia has launched clean city standard indicators, proposed sectoral green city indicators, and adapted one target of global sustainable development goal 11 (UN SDG 11), to improve its urban quality and sustainability. However, using these indicators is not sufficient towards achieving urban sustainability because these indicators are limited in social and economic dimensions. Hence, this study aims to develop all dimensional indicators of sustainability based on all targets of UN SDG 11 with the above indicators. This study focused on the priorities of indicators in Cambodia verified and prioritized by Delphi and analytic hierarchy process (AHP) techniques. Then, a priority-based urban sustainability index for Cambodia was formed based on the concept of sustainability in developing countries. Finally, the standard scores were applied to comparatively assess the sustainability of capital and emerging secondary cities of Cambodia based on the 2018 Commune Database. Through this application, the study also sought to find out whether the priority weights of indicators are necessary for the comparative assessment. The results showed that the sustainability levels of Phnom Penh and Sihanoukville were found to be strong in all environmental, social, and economic dimensions. Battambang is also strong although economic sustainability is slightly lower than the average. Siem Reap is low in economic sustainability level while Poi Pet is remarkably low in environmental and social sustainability. Furthermore, the ranks of sustainability levels of the five cities based on weighted scores are different from their ranks based on unweighted scores. Therefore, this study confirms that priority weights of indicators are necessary for the comparative assessment towards improving the accuracy of the comparison.

**Keywords:** sustainability assessment; sustainable development goals; sustainability in developing countries; sustainable cities; Cambodia; Delphi; AHP; Z-score; comparative assessment

## 1. Introduction

The world is rapidly urbanizing. Currently, more than half of its population lives in urban areas [1]. As projected by the United Nations, 68% of the world population is expected to live in urban areas by 2050, with close to 90% of this increase taking place in Africa and Asia [2]. Similarly, the urban population in Southeast Asia, as projected by the Martin Prosperity Institute, is expected to grow from 280 million in 2017 to 373 million in 2030 [3]. Therefore, improving the sustainable development of cities is necessary for all countries across the world, particularly the developing countries in Southeast Asia like Cambodia.

In order to improve urban quality and sustainability, Cambodia has put many efforts as seen in various mechanisms. In 2011, Cambodia established a clean city assessment committee and approved the clean city standard identified assessment criteria and indicators [4]. This standard's criteria and indicators were used to assess Cambodian cities through a clean city contest every three years. The winners (cities) are awarded by the Prime Minister in the following three names 'Clean City Romduol I, II, and III' upon the winning score [5–7]. The first clean city contest was organized in 2015, and 35 khans and cities were awarded [8]. The second clean city contest was organized in 2018, and 38 khans and cities were awarded [9]. The purposes of the contest are (a) improving livelihood and living environments in cities; (b) improving images of all cities in Cambodia; (c) promoting green services in cities; (d) protecting and promoting urban environments; (e) strengthening and developing urban economies; (f) promoting social equity in cities, especially poverty reduction through tourism: expanding the flow of tourists to all regions in Cambodia [6].

In 2015, the government established a sustainable development council to promote sustainable development in Cambodia [10], and developing sustainable cities is a key policy priority for this council [11]. This council has developed the Code for Environment and Natural Resources, and one chapter of the Code titled "sustainable cities" (Chapter 4) [12] is specifically to deal with the United Nations sustainable development goal 11 (global SDG 11) "Make cities and human settlements inclusive, safe, resilient, and sustainable" [13,14]. This council also collaborated with a Global Green Growth Institute on the Cambodian green urban development program. The goal is to develop cities to be clean, green, and competitive while offering a safe and quality lifestyle to its residents [11,15]. This program approved a green city strategic planning methodology for Cambodia, produced a green city strategic plan 2017–2026 with a list of priority green city investment projects for Phnom Penh, and proposed sectoral objectives and indicators for green city development.

As the sustainable development goals (SDGs) is a global agenda for developing the world, a member state of the United Nations, and the need to contribute to the development and protection of the earth, Cambodia accordingly adapted the SDGs. The Ministry of Planning was delegated by the government to lead and facilitate the SDGs localization into the Cambodian context [16]. In November 2015, this ministry started a meeting to introduce and review the list of global SDGs indicators. From June to October 2016, this ministry conducted interministerial meetings to select indicators and define indicators' definition, as well as a method to calculate the value of indicators. In May and September 2017, this ministry further conducted interministerial meetings to discuss the draft of the Cambodian SDGs (CSDGs) framework [17]. As a result, the government approved the CSDGs framework with identified 18 goals, 88 targets, and 148 indicators. The CSDG 11, named sustainable cities, adapted one target of the global SDG 11 measuring by four indicators [18].

Currently, Cambodia has 28 cities, and Phnom Penh is the capital and largest city with a population of more than two million. According to the World Bank (2018), secondary cities of Cambodia such as Siem Reap, Battambang, Poi Pet, and Sihanoukville are rapidly emerging, with a population of more than 100,000 [19]. Then, using the clean city standard, proposed sectoral green city, and CSDG 11 indicators is not sufficient to assess these capital and emerging cities because the above indicators are limited in social and economic dimensions, particularly in demographic, housing, and employment indicators, which are significant indicators addressed in the UN SDG 11. Therefore, this study aims to develop all dimensional indicators of sustainability to assess these cities based on both UN (SDG 11 indicators) and domestic (CSDG 11 and clean and green city indicators) concepts by focusing on the priorities of assessment indicators in Cambodia.

Furthermore, the current city assessment indicators of Cambodia developed by the National Committee for Clean City Assessment [20], as well as environmentally sustainable cities (ESC) indicators endorsed by the Association of Southeast Asian Nations [21] currently do not use the priority weights of indicators for their city assessments whereas some studies, such as Lee and Lee (2014) [22], Lee (2015) [23], and Han and Lee (2017) [24], applied the priority weights of indicators in the comparative assessment of cities. Therefore, through the application of a priority-based urban sustainability index

to assess Cambodian cities, this study also seeks to find out whether the priority weights of indicators are necessary for the comparison.

## 2. Literature Review

### 2.1. Background of Sustainability

In 1992, the United Nations Conference on Environment and Development, widely known as the **1992 Rio Earth Summit**, was held in Rio de Janeiro, Brazil as a major conference aimed to bring all the United Nations member states together to cooperate internationally on development issues after the Cold War. In total, 172 governments, 108 represented by heads of state or government, adopted three major agreements to guide future approaches to development "Agenda 21", a global plan of action to promote sustainable development. The 1992 Rio Declaration on Environment and Development was a series of principles defining the rights and responsibilities of states. It set out the Statement of Forest Principles: A set of principles to underpin the sustainable management of forests worldwide. Two legally binding instruments were also opened for signature: The United Nations Framework Convention on Climate Change and the Convention on Biodiversity [25].

In 2002, the United Nations World Summit on Sustainable Development, widely known as the **Rio + 10 Earth Summit**, was held in Johannesburg, South Africa. This Summit was an after-10-year discussion, presenting an exciting opportunity for the leaders to adopt concrete steps and identify quantifiable targets for better implementing Agenda 21 [26]. Again, it brought together leaders from governments, businesses, and NGOs to agree on a range of measures toward similar goals, and sustainable development was recognized as an overarching goal for institutions at the national, regional, and international levels [27]. The Rio+10 Summit provided a sustainable development action plan that highlighted the need to enhance the integration of sustainable development in the activities of all relevant United Nations agencies, programs, and funds [28].

In 2012, the United Nations Conference on Sustainable Development, widely known as the **Rio + 20 Earth Summit**, was held in Rio de Janeiro, Brazil. This Sustainable Development Summit was a follow-up to the 1992 Rio Earth Summit and the 2002 Rio+10 Earth Summit. In this Summit, the United Nations member states decided to launch a process to develop a set of sustainable development goals, which is an improvement of millennium development goals and convergence with the post-2015 development agenda and adopted ground-breaking guidelines on green economy policies. Moreover, they decided to establish an intergovernmental process under the General Assembly to prepare options on a strategy for sustainable development financing. They also agreed to strengthen the United Nations Environment Program on several fronts with action to be taken during the 67th session of the General Assembly. Importantly, they agreed to establish a high-level political forum for sustainable development [28,29].

### 2.2. Sustainability in Developing Countries

The World Commission on Environment and Development (1987) [30] proposed the most consensual definition of sustainable development that "The development that meets the needs of the present without compromising the ability of future generations to meet their own needs".

Over the past three decades, the scope of sustainable development has been expanded to include public health, population growth and migration, and poverty reduction worldwide [31]. Although the research on sustainable development has increasingly focused on environmental conservation and natural resource management [32,33], there is still little consideration for the vulnerability of urbanization, particularly for the case of developing countries, where both phenomena persist and adversely affect ecological and environmental resources [34].

There are case studies in developing countries increasingly reported on the failures of sustainable development to achieve a good convergence between economic and social environment [35,36]. According to Cobbinah et al. (2015) [33], Alkire and Santos (2011) [37], and Chandy et al. (2013) [38],

Africa has been identified as the only region where the incidence of poverty has been increasing, with the continent's share of global poverty expected to reach 82% in 2030. Meanwhile, UN official statistics indicate that developing countries are expected to experience an unprecedented rate of urbanization in the foreseeable future amidst persistent poverty conditions [39].

A variety of contexts in developing countries has focused on urbanization and sustainable development in several studies [32,40]. In particular, the definition of sustainable development is based on the assumption that economic development in some ways creates adverse impacts on the environment and ecological systems. Consequently, poverty reduction requires natural resources and environmental management. However, in developing countries, environment and natural resource management have apparently not yet been given high priority. This is consistent with the findings of Visser (2008) [41] who argues that in developing countries, social issues are generally given more political and economic and media emphasis than environmental issues. Although having a balance between poverty reduction and environmental protection, particularly in the context of developing countries is a major challenge, research indicates it is possible and in particular, entrepreneurial actions have the potential to preserve ecosystems, prevent climate change, and reduce environmental degradation and pollution [42,43].

The above studies prove that in the situation of developing countries, environmental issues have apparently not yet been given high priority. Therefore, achieving sustainability in developing countries needs more considerations on environmental issues (see Figure 1).

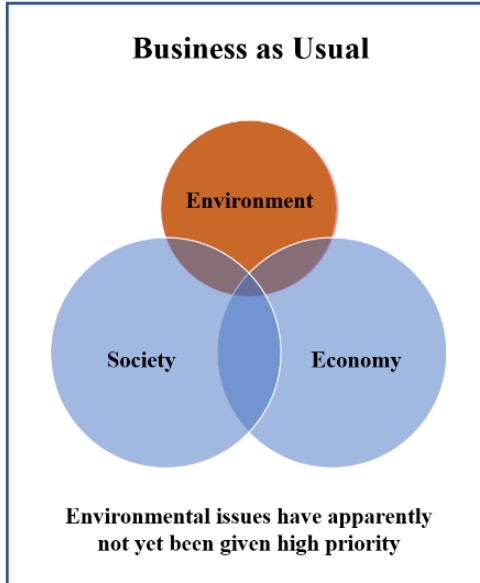 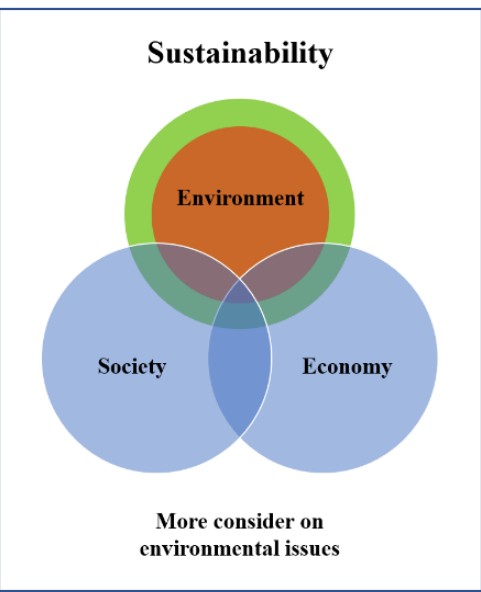

**Figure 1.** Sustainability in developing countries.

Consequently, there are growing interests in promoting green economic or low-carbon development and green growth in developing countries, especially in Asia and the Pacific [44]. Consequently, a Global Green Growth Institute (GGGI) was established as an international green growth organization during the United Nations Conference on Sustainable Development in Brazil (The Rio + 20 Earth Summit in 2012) [45]. This organization has an important mission to promote sustainable development of developing countries by: (a) supporting and diffusing a new paradigm of economic growth "green growth" which is a balanced advance of economic growth and environmental sustainability; (b) targeting key aspects of economic performance and resilience, poverty reduction, green job creation, and social inclusion, and those of environmental sustainability, such as climate change mitigation and adaptation, biodiversity protection and securing access to affordable, clean energy, clean water, and clean land; and (c) creating and improving the economic, environmental and social conditions

of developing and emerging countries through partnerships between developed and developing countries and the public and private sectors [46].

### 2.3. Sustainable City Definitions

Experts, as well as the dictionary, have yet to define what a sustainable city is; however, the general consensus is that a sustainable city is a city designed with consideration for social, economic, environmental impact, and resilient habitat for existing populations, without compromising the ability of future generations to experience the same. According to the United Nations Development Program (UNDP), being a "sustainable city" means "investment in public transport, creating green public spaces, and urban planning and management" [47,48].

Based on the sustainable development concept, sustainability comprises three major dimensions "Environment, Society, and Economy". These dimensions are sometimes called "Planet, People, and Profit" (see Figure 2). Furthermore, the sustainability aspect also includes overlapping dimensions, called livable, equitable, and viable. "Livable" is the correspondence of the environment to social needs. "Equitable" is the interaction between the economic and social dimensions. "Viable" is that economic development must abide by the supportive capacity of the ecosystems, and depletion of nonrenewable resources must be avoided [49–51]. Therefore, sustainable city development aims at achieving human well-being, environmental soundness, and stable economic growth in the cities.

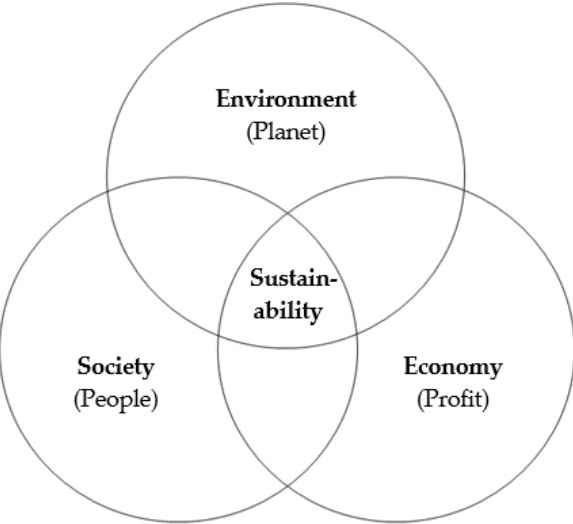

**Figure 2.** Sustainability dimensions.

Likewise, sustainable city development in Cambodia, particularly in Phnom Penh capital, aims at decoupling economic growth from environmental impacts, increasing social inclusion, reducing poverty levels, improving urban welfare, providing urban resilience for all citizens to natural, climatic and other risks, and ensuring urban competitiveness and attractiveness to businesses [52,53]. Towards achieving sustainable city development, the Cambodian National Council for Sustainable Development currently focuses on eight key urban sectors as shown in Figure 3.

Moreover, the Cambodian sustainable city development plan is accompanied by the green city strategic planning methodology approved in 2016 [11], which is a step-by-step guide for municipalities, district and commune officials, and the relevant stakeholders of each municipality across Cambodia seeking to embark on the process of transforming their cities towards green growth. The strategic planning methodology supports cities to assess and prioritize green growth options (investment projects and policy reforms) for implementation. This methodology holistically considers all aspects of green urban development, such as low-carbon development, climate resilience, resource efficiency, as well as social inclusion and poverty alleviation [53].

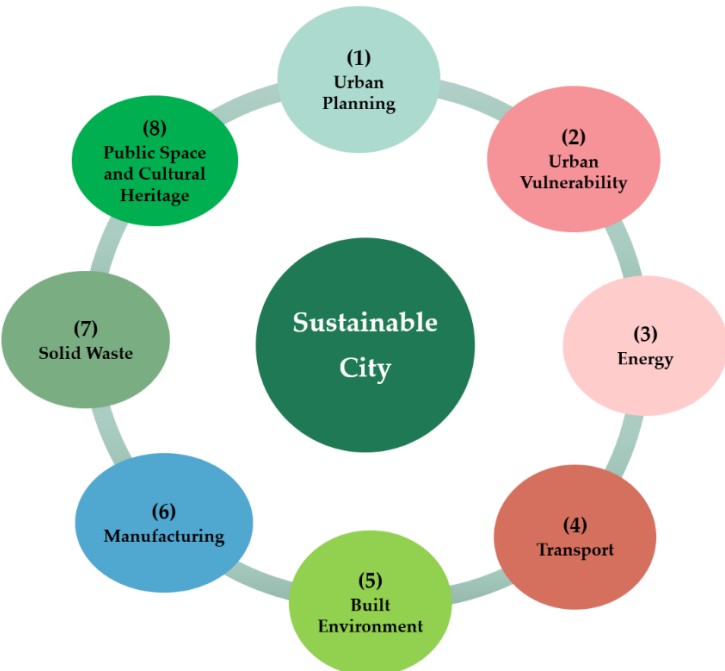

**Figure 3.** Key sectors for sustainable city development in Cambodia. Source: National Council for Sustainable Development (NCSD) and Global Green Growth Institute (GGGI) 2019, drawn by the author.

*2.4. Global Sustainable City Goal, Targets and Indicators*

By recognizing the success of the implementation of the millennium development goals (MDGs) and as needed a new development agenda beyond 2015, the United Nations Conference on Sustainable Development in 2012 (Rio+20 Earth Summit) under the theme "**The Future We Want**" agreed to establish an open working group to develop a set of sustainable development goals by improving the eight MDGs and their 21 targets.

After negotiations for more than one year, the open working group proposed the 17 goals as the sustainable development goals [13]. These sustainable development goals (SDGs) were officially announced in 2015 at the United Nations Headquarters. These SDGs were adopted by the world leaders in September 2015 and came to force on 1 January 2016 [54,55].

To sustain the rapid urbanization across the world, one of the sustainable development goals (SDG 11) aims to make cities and human settlements inclusive, safe, resilient, and sustainable, widely known as the global "sustainable cities" goal. The global SDG 11 identified ten targets, and 15 assessment indicators as summarized in Table 1 [56,57].

**Table 1.** United Nations sustainable development goal 11 (SDG 11) targets and indicators.

| | Target | | Indicator |
|---|---|---|---|
| 11.1 | Ensuring adequate and affordable housing for all by 2030 | 1 | Proportion of urban population living in slums and poor/informal settlements |
| 11.2 | Providing safe and affordable transport systems for all by 2030 | 2 | Proportion of population convenient access to public transport for everyone in the city |
| 11.3 | Enhancing inclusive urbanization and participatory human settlement planning by 2030 | 3 | Ratio of land consumption to population growth |
| | | 4 | Proportion of cities with participation structure of civil society in urban planning |

**Table 1.** *Cont.*

| | Target | | Indicator |
|---|---|---|---|
| 11.4 | Strengthening efforts to protect cultural and natural heritages | 5 | Total expenditure spent on protection and conservation of cultural and natural heritages |
| 11.5 | Reducing the number of people affected by disasters with a focus on protecting the poor by 2030 | 6 | Number of people affected by disasters |
| | | 7 | Direct economic loss in relation to global GDP, damage to critical infrastructure |
| 11.6 | Reducing the environmental impact by paying attention to air quality and waste management by 2030 | 8 | Proportion of solid waste regularly collected |
| | | 9 | Annual mean levels of fine particulate matter (e.g., PM2.5 and PM10) in cities |
| 11.7 | Providing universal access to urban green and public spaces by 2030 | 10 | Average share of open space for all |
| | | 11 | Rate of persons victims of physical harassment |
| 11.a | Supporting positive links between urban, periurban, and rural areas | 12 | Proportion of population living in cities that integrate population projections and resource needs |
| 11.b | Increasing the number of cities adopting integrated policies and plans towards inclusion by 2020 | 13 | No. of countries adopted disaster risk reduction strategy |
| | | 14 | Proportion of local governments that adopt/implement local disaster risk reduction strategies |
| 11.c | Supporting least developed countries in constructing resilient buildings | 15 | Proportion of financial support to the least developed countries for buildings utilizing local materials |

**Source:** United Nations (2015) and UN-Habitat (2016), summarized by the author.

*2.5. Cambodian Sustainable City Goal, Target, and Indicators*

Cambodian sustainable development goals (CSDGs) framework identified 18 goals, 88 targets, and 148 indicators which were approved by the Council of Ministers in the full cabinet meeting in November 2018 [18]. The most striking difference of CSDGs framework from the global SDGs framework is the adoption of an additional CSDG 18 "demining and explosive remnants of war (ERW)". There are important changes to the targets and indicators as well. Referring firstly to the targets, while the target statements have been retained in full, there has been an overall reduction, and adoption has varied between goals according to applicability. As shown in Figure 4, the level of variation is more marked on some goals. In particular, the number of targets is lower in Goals 8 to 12 and 14. Especially, CSDG 11 adopted only one target of the global SDG 11.

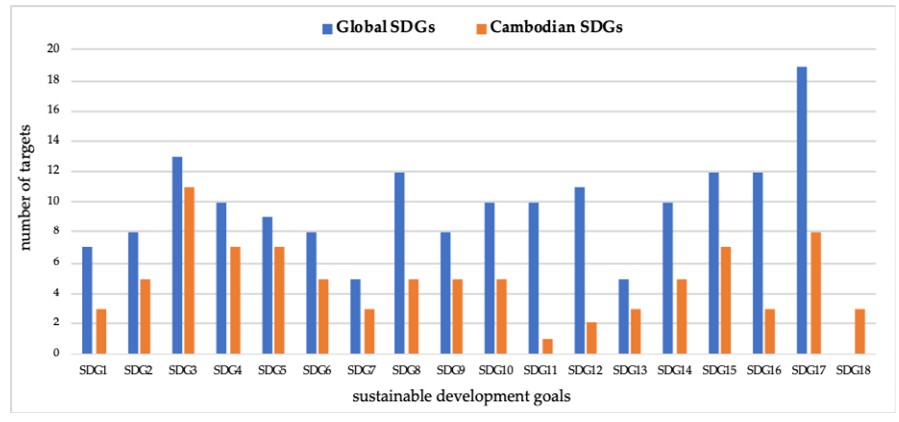

**Figure 4.** Number of United Nations and Cambodian SDGs targets (by goal). Source: Royal Government of Cambodia (RGC) 2018, graphed by the author.

The adapted SDG 11 target is "By 2030, reduce adverse per capita environmental impact of cities, including by paying special attention to air quality and municipal and other waste management" [18]. This target is being measured by four indicators as shown in Table 2. These indicators focused on air quality and waste management responsible by the Ministry of Environment.

**Table 2.** Cambodian SDG 11 target and indicators.

| Target | Indicator |
|---|---|
| By 2030, reduce adverse per capita environmental impact of cities, including by paying special attention to air quality and municipal waste management | Amount of urban solid waste regularly collected and with adequate final discharge |
| | Percentage of the deduction plastic bag used |
| | Percentage of the solid wasted segregate by technical guidance |
| | The number of sites monitored by the air quality on the parameter is the annual average of parameters of CO, $NO_2$, $SO_2$, TSP, PM2.5 and PM10 |

The priority of CSDG 11 is lower compared to other CSDGs as Cambodian cities are not yet highly populated. Thus, developing cities is generally considered similar to the development of districts. However, the capital and largest city, Phnom Penh, has high density, and four secondary cities (Siem Reap, Battambang, Poi Pet, and Sihanoukville) are rapidly emerging. Therefore, the four CSDG 11 indicators are not sufficient for assessing these cities towards achieving urban sustainability because these indicators lack social and economic dimensions.

*2.6. Cambodian Clean and Green City Indicators*

Cambodia has clean city standard identified assessment criteria and indicators to monitor and assess Cambodian cities through a national clean city contest every three years. The clean city standard indicators were classified into eight categories: (1) environmental management, (2) cleanliness, (3) waste management, (4) raising awareness on environmental protection and cleanliness, (5) green spaces, (6) urban safety and security, and (7) tourism infrastructure [6]. These standard indicators focused on urban cleanliness and tourism infrastructures and facilities because these indicators were developed by the National Committee for Clean City Assessment and the Ministry of Tourism, aiming to promote tourism in all cities across Cambodia [7].

The clean city standard indicators are limited in social and economic dimensions; however, currently, these indicators are still important, and the government is still using these standard indicators to assess the urban development progress and the quality of cities in Cambodia. This is because most cities in Cambodia are not highly populated (see Table 3). Then, the development of cities in Cambodia, except the capital city, is generally considered similar to the development of districts. Though, the secondary cities, such as Siem Reap, Battambang, Poi Pet, and Sihanoukville are rapidly emerging, with the population above 100,000 [19], especially the Phnom Penh capital has high density; therefore, using the current clean city indicators is not sufficient in assessing these capital and emerging secondary cities towards achieving urban sustainability.

**Table 3.** List of Cambodian cities from high to low population.

| No. | City | Status | Population | Source |
|-----|------|--------|-----------|--------|
| 1 | Phnom Penh | Capital of the country | 2,129,371 | [58] |
| 2 | Siem Reap | Municipality of Siem Reap province | 250,798 | [59] |
| 3 | Battambang | Municipality of Battambang province | 163,347 | [60] |
| 4 | Poi Pet | A city in Banteay Meanchey province | 104,156 | [61] |
| 5 | Sihanoukville | Municipality of Preah Sihanouk province | 100,856 | [19,62] |
| 6 | Serei Saophoan | Municipality of Banteay Meanchey province | 87,738 | [61] |
| 7 | Ta Khmau | Municipality of Kandal Province | 82,671 | [63] |
| 8 | Pursat | Municipality of Pursat Province | 68,247 | [64] |
| 9 | Samraong | Municipality of Oddar Meanchey province | 64,639 | [65] |
| 10 | Stung Saen | Municipality of Kampong Thom province | 59,044 | [66] |
| 11 | Chbar Mon | Municipality of Kampong Speu province | 50,252 | [67] |
| 12 | Svay Rieng | Municipality of Svay Rieng province | 47,829 | [68] |
| 13 | Doun Kaev | Municipality of Takeo province | 45,446 | [69] |
| 14 | Bavet | A city in Svay Rieng province | 42,546 | [68] |
| 15 | Kaeb | Municipality of Kaeb province | 41,818 | [70] |
| 16 | Kampong Chhnang | Municipality of Kampong Chhnang province | 40,911 | [71] |
| 17 | Kampong Cham | Municipality of Kampong Cham province | 40,015 | [72] |
| 18 | Suong | A city in Tboung Khmum province | 39,452 | [73] |
| 19 | Kampot | Municipality of Kampot province | 38,595 | [74] |
| 20 | Pailin | Municipality of Pailin province | 34,948 | [75] |
| 21 | Stung Treng | Municipality of Stung Treng province | 33,063 | [76] |
| 22 | Banlung | Municipality of Ratanakiri province | 32,294 | [77] |
| 23 | Khemarak Phoumin | Municipality of Koh Kong province | 30,894 | [78] |
| 24 | Kratie | Municipality of Kratie province | 30,435 | [79] |
| 25 | Prey Veng | Municipality of Prey Veng province | 26,011 | [80] |
| 26 | Preah Vihear | Municipality of Preah Vihear province | 22,983 | [81] |
| 27 | Sen Monorom | Municipality of Mondulkiri province | 13,871 | [82] |
| 28 | Koh Rong | A city in Preah Sihanouk province | - | [83] |

**Note:** Koh Rong (No. 28) is a new coastal city formed in 2019. This newest city does not have official data on its population yet; however, according to the Khmer Times newspaper, permanent residents of this city are around 4000 [84].

Furthermore, Cambodia also proposed green city sectoral objectives and indicators through a green urban development program. Since 2015, the Cambodian government through its National Council for Sustainable Development has been collaborating with the Global Green Growth Institute on this program [85]. Through various national consultation workshops and technical meetings, the green city development framework for Cambodia was defined, and a green city strategic planning methodology was approved in 2016. The green city strategic plan 2017–2026 and the list of priority green city investment projects for Phnom Penh were also proposed [11,15]. The goal is to develop cities to be clean, green, and competitive while offering a safe and quality lifestyle to its residents.

The green city indicators were proposed along with their sectoral objectives and classified by eight key urban sectors: (1) urban planning, (2) urban vulnerability, (3) energy, (4) transport, (5) built environment, (6) manufacturing, (7) solid waste, and (8) public space and cultural heritage. However, these proposed green city indicators have never been used and applied to assess the development and

management of green cities since development. The proposed indicators were also not clear in their measuring units. Similar to clean city indicators, these indicators are limited in social and economic dimensions, particularly in demographic, housing, and safety indicators.

*2.7. Sustainability Assessment of Cities*

In terms of assessment, indicators are widely known as an important tool not only in assessing the progress of sustainable development but also in making the concept appealing to a wide range of potential stakeholders [86–88]. Likewise, the indicators are found to be powerful tools for making environmental and social dimensions important enabling their management although assessing the impact of the indicators on policymaking and progress towards sustainability is usually difficult [89]. This is because sustainability is a cross-cutting aspect that comprises all economic, social, and environmental dimensions, particularly the overlapping dimensions, livable, viable, and equitable [49,90]. In developing countries, achieving economic and social dimensions are commonly given higher priority than the environmental dimension while economic and social development generally has negative impacts on environments. The negative impacts are seen in both developed and developing countries across the world. Therefore, environmentally sustainable (eco or green) city assessment is getting more attention, particularly in Asian countries.

The Association of Southeast Asian Nations (ASEAN; Brunei, Cambodia, Indonesia, Laos, Malaysia, Myanmar, Philippines, Singapore, Thailand, and Vietnam), developed environmentally sustainable city (ASEAN ESC) indicators [91]. The goal is to pursue environmental sustainability in the rapidly growing cities of ASEAN member countries [92]. Furthermore, ASEAN clean tourist city standard was also approved in 2016. The goal is to provide ASEAN member countries with a tool that will improve the quality of tourism in their cities, increase their marketing competitiveness but also improve the situation of local residents and livelihood by alleviating poverty [93]. Likewise, South Korea, a developed country in Asia, is growing interest in promoting urban regeneration towards achieving urban sustainability [94], as well as the New Deal policy. The President Moon Administration declared to conduct the "Urban Renewal New Deal Project" as a policy project to overcome the problems in the new town-oriented development and to revitalize the old downtown and residential areas [95,96]. Consequently, there was a study recently assessing the sustainability of South Korean cities. The study focused on the resilience of sustainable urban development in 72 cities. The results of this study found that the sustainability level of Seoul is low in forest resources and rent housings while Busan is weak in unemployment and economic activity participation [97,98]. This shows that even South Korea, an advanced country, still needs an efficient sustainability assessment tool for guiding the development and management of its cities. This also proves that sustainability assessment tools are very significant to measure how the cities are progressing and to direct their development progress towards achieving sustainability. In particular, the sustainability assessment can reveal the weak points of cities that need to be improved.

## 3. Materials and Methods

*3.1. Urban Sustainability Index*

To assess the sustainability of the capital and emerging secondary cities of Cambodia, this study developed an assessment index based on the United Nations (global SDG 11 indicators) and domestic (CSDG 11 and clean and green city indicators) concepts by focusing on the importance (priorities) of the developed urban sustainability indicators in Cambodia. Furthermore, achieving sustainability in developing countries needs more consideration on environmental issues and the balance of environmental, social, and economic dimensions. Therefore, the developed priority-based urban sustainability assessment index must contain equally the number of environmental, social, and economic indicators. Hence, this study accordingly rounded up the number of three-dimensional sustainability indicators based on the priority weight.

The urban sustainability assessment indicators for Cambodia were developed and prioritized by the first [99] and second steps [100] of this research. As shown in Table 4, there are 32 consensus urban sustainability assessment indicators listed from high to low priority. These indicators were developed by using the consensus method, namely Delphi. The initial development of indicators (Delphi panel round one) was based on UN SDG 11, ASEAN ESC, clean and green city (CC and GC), and relevant case study indicators. Some indicators were provided by 16 experienced panelists who are the practitioners in the fields of sustainability. Then, the initial indicators were validated in Delphi round two and three. Finally, 32 indicators were obtained after consensus analysis.

**Table 4.** Prioritized consensus urban sustainability assessment indicators for Cambodia.

| Rank | Indicator (In-short) | Weight | CC | GC | UN SDG 11 | CSDG 11 |
|------|----------------------|--------|----|----|-----------|---------|
| 1 | Slum-poor settlement rate | 0.0557 | | | √ | |
| 2 | Unemployment rate | 0.0516 | | | √ | |
| 3 | Crime prevention | 0.0470 | √ | | | |
| 4 | Potable water supply | 0.0469 | | | √ | |
| 5 | Population growth rate | 0.0462 | | | √ | |
| 6 | Low-income housing ratio | 0.0445 | | | | |
| 7 | Solid waste collection rate | 0.0437 | √ | √ | √ | √ |
| 8 | Labor force participation rate | 0.0421 | | | √ | |
| 9 | Construction risk prevention | 0.0400 | | | | |
| 10 | Traffic congestion reduction | 0.0398 | | | | |
| 11 | Average household income | 0.0366 | | | | |
| 12 | Water reservoir coverage | 0.0362 | | √ | | |
| 13 | Quality residential buildings | 0.0356 | | | | |
| 14 | Wastewater treatment ratio | 0.0352 | √ | √ | | |
| 15 | New jobs created per year | 0.0345 | | | | |
| 16 | Water consumption rate | 0.0320 | √ | | | |
| 17 | Urban population density | 0.0291 | | | | |
| 18 | Energy consumption rate | 0.0284 | | √ | | |
| 19 | Waste reduction initiatives | 0.0284 | | | | |
| 20 | Disaster prevention | 0.0274 | | | √ | |
| 21 | Public transport sharing rate | 0.0268 | √ | √ | √ | |
| 22 | Insurance-registered ratio | 0.0237 | | | | |
| 23 | Sidewalk improvement | 0.0234 | | | | |
| 24 | Urban park ratio | 0.0229 | √ | √ | | |
| 25 | No. of public parking lots | 0.0209 | | | | |
| 26 | Heritage conservation budget | 0.0195 | | √ | √ | |
| 27 | Renewable energy shared rate | 0.0178 | √ | √ | | |
| 28 | Tourism growth rate per year | 0.0173 | √ | | | |
| 29 | Urban forest ratio per capita | 0.0150 | | | | |
| 30 | Fine dust level (PM2.5, PM10) | 0.0124 | | | √ | √ |
| 31 | Number of playgrounds | 0.0114 | | | | |
| 32 | Biodiversity gardens | 0.0086 | | | | |
| | **Total** | **1.0000** | | | - | |

All 32 consensus indicators were then prioritized by using the pairwise comparison method, namely the analytic hierarchy process (AHP). A total of 102 consistent AHP questionnaires were gathered from offline (face-to-face interviews—used mostly with government officials) and online

(Email, Facebook, and LinkedIn) surveys. The surveyed respondents are the potentials working and experienced in the fields of urban planning, development, management, and assessment in Cambodia. The results of the prioritization, priority weights, are shown in the table above.

This study selected only the prioritized consensus urban sustainability assessment indicators of UN SDG 11, Cambodian SDG 11, and clean and green city (CC and GC) indicators. The selected consensus urban sustainability indicators are donated by the sign "√".

According to the above table, there are 18 consensus urban sustainability assessment indicators for Cambodia based on United Nations (UN SDG 11) and Cambodian (CSDG 11, CC, and GC) contexts. These 18 indicators listed from high to low priority are shown in Table 5. The sustainability dimensions of these indicators were identified in this table as well.

**Table 5.** Urban sustainability indicators selected based on United Nations and Cambodian contexts.

| Rank | Indicator | Weight | Sustainability Dimension |
|---|---|---|---|
| 1 | Slum-poor settlement rate | 0.0557 | Social |
| 2 | Unemployment rate | 0.0516 | Economic |
| 3 | Crime prevention | 0.0470 | Social |
| 4 | Potable water supply ratio | 0.0469 | Social |
| 5 | Population growth rate | 0.0462 | Economic |
| 6 | Solid waste collection rate | 0.0437 | Environmental |
| 7 | Labor force participation rate | 0.0421 | Economic |
| 8 | Water reservoir coverage | 0.0362 | Economic |
| 9 | Wastewater treatment ratio | 0.0352 | Environmental |
| 10 | Water consumption rate | 0.0320 | Social |
| 11 | Energy consumption rate | 0.0284 | Economic |
| 12 | Disaster prevention | 0.0274 | Social |
| 13 | Public transport sharing rate | 0.0268 | Environmental |
| 14 | Urban park ratio | 0.0229 | Social |
| 15 | Heritage conservation budget | 0.0195 | Social |
| 16 | Renewable energy shared rate | 0.0178 | Environmental |
| 17 | Tourism growth rate per year | 0.0173 | Economic |
| 18 | Fine dust level (PM2.5, PM10) | 0.0124 | Environmental |

In fact, each of the sustainability–dimensional indicators generally targets to achieve more than one aspect of sustainability. For example, the slum-poor settlement rate targets achieve both social and environmental dimensions, which means slums and poor settlements contain both social (poor urban housings and settlements) and environmental (poor urban residential environments) aspects. Hence, this study determines the dimension that the indicator targeted most is the dimension of that indicator. Consequently, the slum-poor settlement rate mostly targets to achieve social dimension (adequate housings in the city); thus, this indicator is defined as a sustainability indicator for social dimension. Furthermore, the energy consumption rate (the ratio of electricity supply available or accessible for households in the city) mostly targets to achieve economic dimension (electricity used for small and medium enterprises, such as textile and wearing apparel, manufacturing of paper product and printing, and manufacturing of wood and wood product [101]). This indicator targets to achieve social dimension as well, but in the city, its social dimension is commonly lower than its economic dimension.

According to the dimensions of urban sustainability indicators identified in the table above, social and economic dimensions are more than environmental dimensions. This is associated with the

statements argued in the literature review that in developing countries, social and economic issues are usually given higher priority than environmental issues. Achieving sustainability in developing countries needs more considerations on environmental issues.

Based on these statements, especially the sustainability principle (balancing environmental, social, and economic dimensions), this study accordingly reduced the low-priority (less important) indicators of social and economic dimensions towards balancing the priority-based consensus urban sustainability indicators for the comparative assessment of cities in Cambodia. Although the number of indicators was rounded up for balancing environmental, social, and economic dimensions, still the total priorities (weights) of the indicators in each dimension are not equal. The environmental indicators seem to have a total weight lower than social and economic dimensions. This shows that in a developing country like Cambodia, urban sustainability assessment indicators for social and economic dimensions are still more important than the environmental dimension because it is the nature of the development in a developing country in order to provide sufficient social needs and good economic conditions to its growing households. Hence, this study just rounded up the number of indicators for each sustainability dimension, not the total priority weight of the indicators.

After round-up the number of indicators by sustainability dimensions, this study accordingly converted the weight of obtained consensus urban sustainability indicators into a 100-scale. How these weights of indicators are converted is shown in Appendix A. The results of the priority-based urban sustainability assessment index after rounding-up the indicators in each sustainability dimension and conversion of the weights into a 100-scale are shown in Table 6.

**Table 6.** Development of priority-based urban sustainability index for Cambodia.

| Dimension | No. | Indicator | Weight | In Percentage |
|---|---|---|---|---|
| Environmental | 1 | Solid waste collection rate | 0.0737 | 7.37% |
| | 2 | Wastewater treatment ratio | 0.0652 | 6.52% |
| | 3 | Public transport sharing rate | 0.0568 | 5.68% |
| | 4 | Renewable energy shared rate | 0.0478 | 4.78% |
| | 5 | Fine dust level (PM2.5, PM10) | 0.0424 | 4.24% |
| Social | 1 | Slum-poor settlement rate | 0.0857 | 8.57% |
| | 2 | Crime rate or prevention | 0.0770 | 7.70% |
| | 3 | Potable water supply ratio | 0.0769 | 7.69% |
| | 4 | Water consumption rate | 0.0620 | 6.20% |
| | 5 | Disaster prevention | 0.0574 | 5.74% |
| Economic | 1 | Unemployment rate | 0.0816 | 8.16% |
| | 2 | Population growth rate | 0.0762 | 7.62% |
| | 3 | Labor force participation rate | 0.0721 | 7.21% |
| | 4 | Water reservoir coverage | 0.0662 | 6.62% |
| | 5 | Energy consumption rate | 0.0584 | 5.84% |
| Total | | 15 | 1.0000 | 100.00% |

There are 15 urban sustainability assessment indicators with five indicators for each dimension. The indicators in each dimension listed orderly from high to low priorities. At the environmental dimension, solid waste collection rate is the highest priority indicator, followed by wastewater treatment ratio and public transport sharing rate. At the social dimension, the slum-poor settlement rate is the highest priority indicator, followed by crime rate or prevention and potable water supply ratio. At the economic dimension, the unemployment rate is the highest priority indicator, followed by population growth rate and labor force participation rate.

### 3.2. City Selection and Locations

This study selected the capital and emerging secondary cities of Cambodia for the comparative assessment. The Phnom Penh capital is the largest city, with a population of 2,129,371 [58]. According to

the World Bank (2018), emerging secondary cities are Siem Reap, Battambang, Poi Pet, and Sihanoukville, with a population ranging from 100,000 to 200,000 [19]. According to the 2018 Commune Database, Siem Reap is the largest emerging secondary city and has a population more of than 200,000. The exact population of this city is 250,798 [59]. Battambang and Poi Pet cities are the second and third emerging secondary cities, respectively. According to the World Bank, these cities have populations between 100,000 and 200,000. The exact population of Battambang is 163,347 [60] while Poi Pet is 104,156 [61]. Sihanoukville is the smallest emerging secondary city compared to Siem Reap, Battambang, and Poi Pet. According to the World Bank (2018), the population of this city is more than 100,000 [19] and its widely known population is 100, 856 [62].

Figures below show the locations of the five capital and emerging cities of Cambodia. These figures were adapted from Google Earth: https://earth.google.com/web/. As shown in Figure 5, most of the cities are located on the western side of the country. Phnom Penh (1) is located at the cross-cutting point of four rivers, namely Tonle Sap, Upper-Mekong, Lower-Mekong, and Bassac rivers. In Cambodian language, the cross-cutting point is called the "Four-Face River" (see Figure 6).

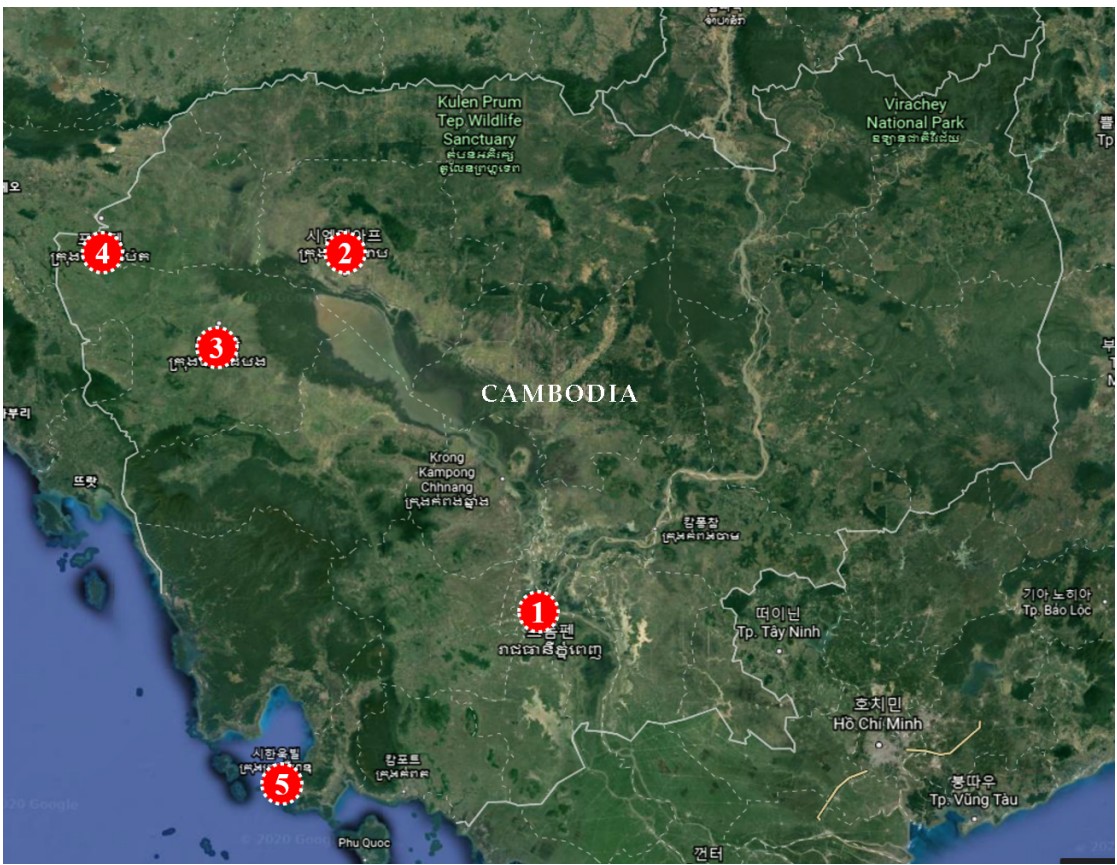

**Figure 5.** Location of five cities in the country's map. Denoted ordinal numbers of cities are listed from high to low population. (**1**) is Phnom Penh. (**2**) is Siem Reap. (**3**) is Battambang. (**4**) is Poi Pet. (**5**) is Sihanoukville. Source: adapted from Google Earth.

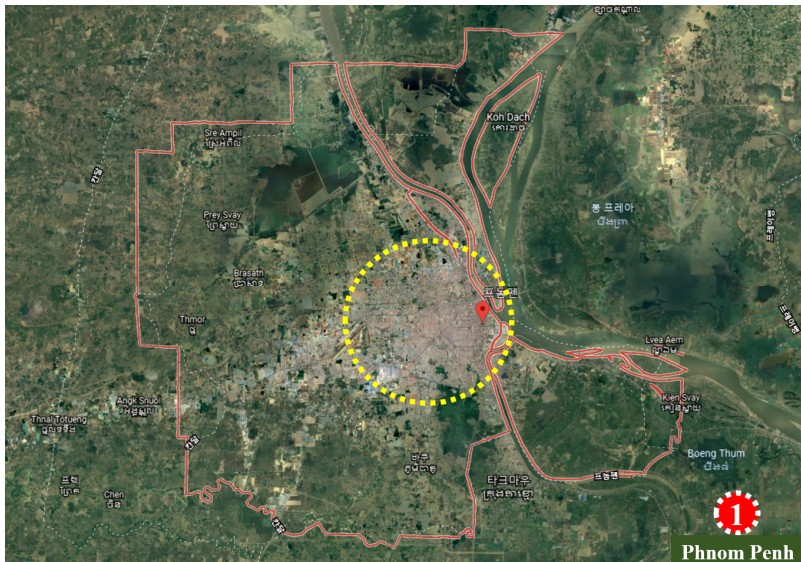

**Figure 6.** Boundary of Phnom Penh. The yellow circle indicates a high-density area. Source: adapted from Google Earth.

Siem Reap (2) is located on the northern side of and above the Tonle Sap Great Lake. This Great Lake is the largest lake in Southeast Asia. This lake is also the most important natural wetland in the region. It covers an area of more than 15,000 km$^2$ [102]. Battambang (3) is located on the northern side of the Tonle Sap Great Lake whereas Poi Pet (4) is located next to the border of Cambodia with Thailand. Finally, Sihanoukville (5), a coastal city, is located next to the sea (see Figure 7).

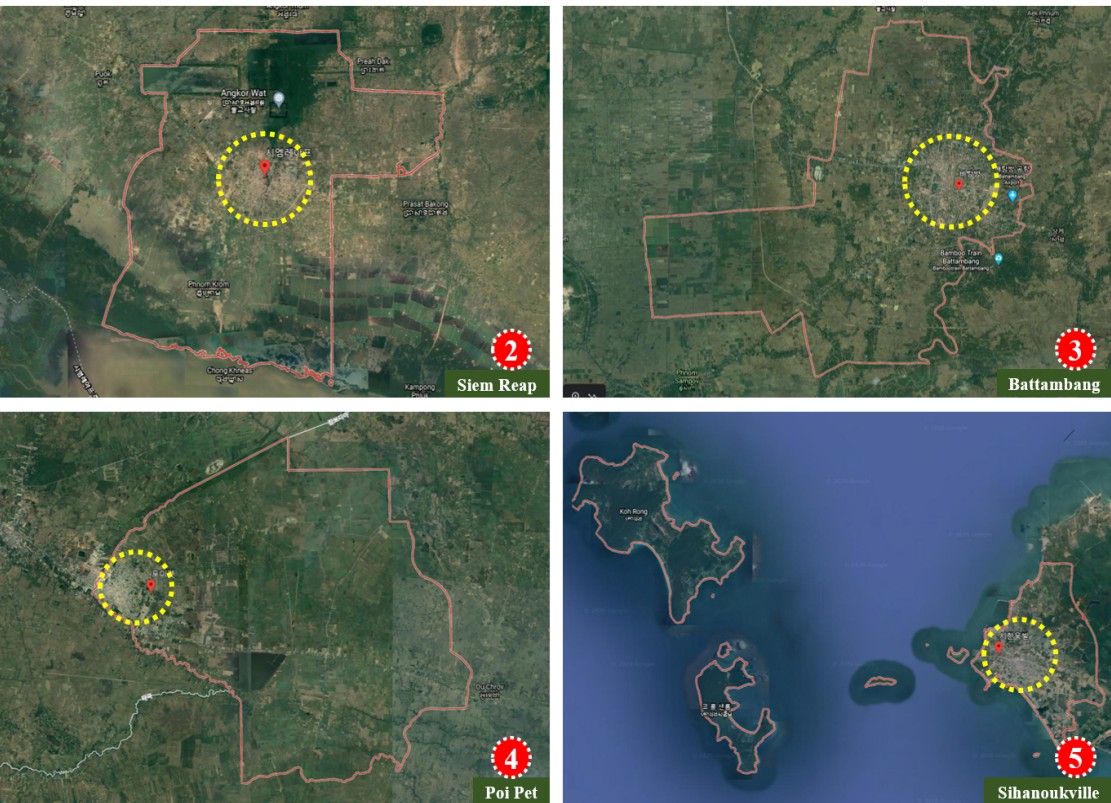

**Figure 7.** Boundary of the four emerging secondary cities: Siem Reap (**2**), Battambang (**3**), Poi Pet, (**4**) and Sihanoukville (**5**). The yellow circles indicate high-density areas. Source: adapted from Google Earth.

### 3.3. Data Sources and Validation

The data of the capital and emerging secondary cities by each indicator were sourced from the "2018 Commune Database", published in early 2019 by the Capital and Provincial Departments of Planning [59–61,103,104] whereas the data of fine dust level (PM2.5) were sourced from the Announcements on "Air Quality Results in the Capital and Provinces of Cambodia" published by the Ministry of Environment (see Appendix B) [105–107]. The data of the Phnom Penh capital were published by the Phnom Penh Capital Department of Planning whereas the data of the Siem Reap city were published by the Siem Reap Provincial Department of Planning. The data of the Battambang city were published by the Battambang Provincial Department of Planning whereas the data of the Poi Pet city were published by the Banteay Meanchey Provincial Department of Planning. Finally, the data of the Sihanoukville were published by the Preah Sihanouk Provincial Department of Planning. These published data are shown with each reference in Table 7.

**Table 7.** Publishing departments of the 2018 Commune Database of five cities and published year.

| City | Publishing Department (Published Year) | Ref. |
|------|----------------------------------------|------|
| Phnom Penh | Phnom Penh Capital Department of Planning (2019) | [103] |
| Siem Reap | Siem Reap Provincial Department of Planning (2019) | [59] |
| Battambang | Battambang Provincial Department of Planning (2019) | [60] |
| Poi Pet | Banteay Meanchey Provincial Department of Planning (2019) | [61] |
| Sihanoukville | Preah Sihanouk Provincial Department of Planning (2019) | [104] |

Therefore, the data used for 15 consensus urban sustainability indicators in this study were sourced from the above commune database of each city, except for one indicator "fine dust level". The data of this indicator were sourced from the Announcements on Air Quality Results measured by the Ministry of Environment. The characteristics of the obtained data are shown in Table 8.

**Table 8.** Characteristics of the data obtained from the 2018 Commune Database.

| Indicator | Characteristics of the Data | Impact |
|-----------|-----------------------------|--------|
| Solid waste | Percentage of households accessing to solid waste collection services | + |
| Wastewater | Percentage of households installed proper toilets/wastewater storages | + |
| Public transport | The ratio of public transport means compared to total transport means | + |
| Renewable energy | Percentage of houses using solar PV | + |
| Fine dust level | Level of PM 2.5 ($\mu g/m^3$) | − |
| Poor housing | Percentage of households living in low-quality housings | − |
| Crime rate | The ratio of murder/robbery/kidnapping cases per 10,000 households | − |
| Potable water supply | The ratio of households accessing to potable water supply infrastructure | + |
| Water consumption | Percentage of households having clean-water consuming habits | + |
| Disaster-affected ratio | The ratio of storms and floods affected per 1000 households | − |
| Unemployment rate | Unemployment rate | − |
| Population growth | Population growth rate per year | + |
| Labor participation | The ratio of the population aged from 18 to 60 years old | + |
| Near water reservoir | Percentage of households accessible to water sources less than 150m | + |
| Energy consumption | Percentage of households accessing to electricity | + |

Among the 15 urban sustainability assessment indicators, there are ten indicators with positive impacts and five indicators with negative impacts. The public transport sharing rate was calculated by the ratio of the number of public transport means compared to the total number of inland transport means. This calculation is shown in Table A2. The fine dust level is based on levels of PM2.5 which were measured and announced by the Ministry of Environment. The calculation of the average level of PM2.5 is shown in Table A3. The disaster-affected ratio is the ratio of households affected by storms and floods per 1000 households. The calculation of this ratio is shown in Table A4. The calculation of the labor force participation rate is shown in Table A5. The calculation of the unemployment rate is

shown in this table as well. The calculated and/or obtained data of all urban sustainability assessment indicators by each city are shown in Table 9.

**Table 9.** The obtained/calculated data of all urban sustainability indicators by each city.

| Indicators | Phnom Penh | Siem Reap | Battambang | Poi Pet | Sihanoukville |
|---|---|---|---|---|---|
| Solid waste | 82.2 | 36.2 | 46.6 | 18.6 | 70.1 |
| Wastewater | 92.5 | 91.1 | 93.7 | 93.4 | 92.8 |
| Public transport | 10.93 | 22.54 | 14.20 | 14.02 | 14.36 |
| Renewable energy | 0.00 | 5.55 | 0.70 | 0.36 | 0.39 |
| Fine dust level | 23.67 | 31.67 | 38.00 | 27.00 | 13.33 |
| Poor housing | 0.08 | 1.20 | 0.00 | 0.20 | 0.10 |
| Crime rate | 0.9 | 0.4 | 0.0 | 0.0 | 6.0 |
| Potable water supply | 98.0 | 98.8 | 85.6 | 83.3 | 97.8 |
| Water consumption | 99.0 | 92.6 | 92.8 | 84.5 | 99.4 |
| Disaster-affected ratio | 6.0 | 0.6 | 1.8 | 1.1 | 0.1 |
| Unemployment rate | 0.38 | 0.14 | 0.32 | 0.28 | 0.00 |
| Population growth | 5.07 | 1.69 | 3.46 | 1.58 | 3.14 |
| Labor participation | 58.47 | 52.63 | 52.18 | 57.01 | 57.34 |
| Near water reservoir | 84.3 | 99.2 | 14.9 | 4182.0 | 92.9 |
| Energy consumption | 99.65 | 98.50 | 98.70 | 99.00 | 99.70 |

These data are used to find out the strengths and weaknesses in sustainability by each city, particularly the strengths and weaknesses in environmental, social, and economic dimensions. In the second national clean city contest in 2018, Siem Reap, Battambang, and Sihanoukville were awarded "Romduol III" (first rank) whereas Poi Pet was awarded "Romduol II" (second rank). Many Khans of the Phnom Penh capital were awarded "Romduol III" and "Romduol II" as well [9]. Furthermore, four of the five cities also received the ASEAN ESC Awards. Therefore, these cities are qualified in environmental aspects based on the national clean city standard and ASEAN environmentally sustainable city indicators. In general, every city has its own unique, strengths, and weaknesses. Consequently, when comparing these qualified cities with each other could find out the strengths, and weaknesses of each city. Hence, this study used the comparative assessment method to find out the strengths and weaknesses of each city. A standard of urban sustainability for Cambodian contexts will also be formed based on this comparative assessment method. The formed urban sustainability standard will be used to reflect the sustainability of each city.

*3.4. Standard Scores for Comparative Assessment*

In statistics, the standard scores or Z-scores, other terms: Z-values, normal scores, and standard variables, is the number of standard deviations where the value of a raw score (for example, values or data observed) is higher or lower than the mean value of what is observed or measured. The above-average raw scores have positive standard scores while the below-average scores have negative standard scores (see Figure 8).

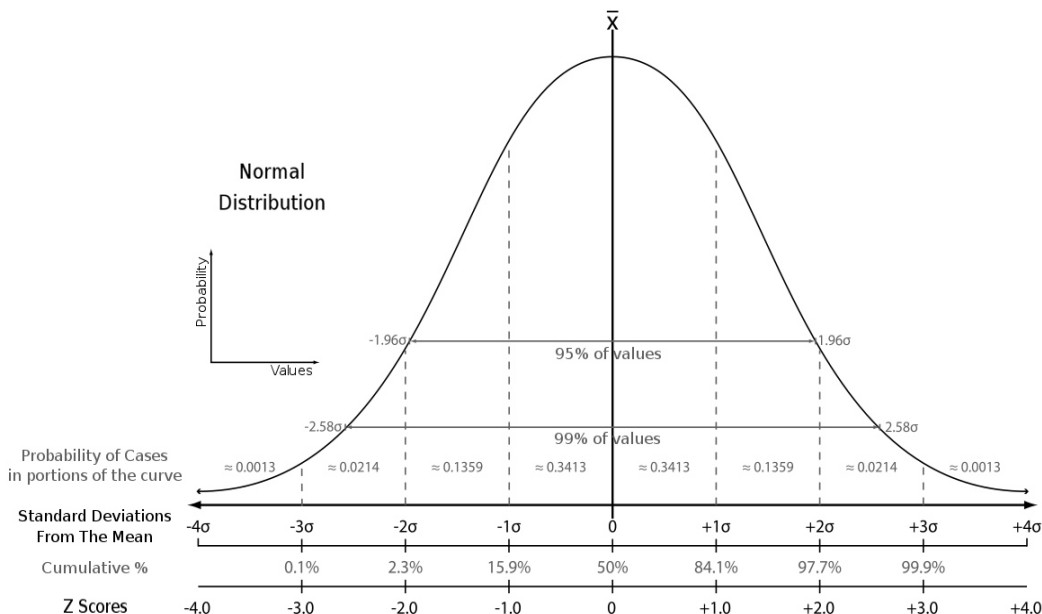

**Figure 8.** Grading methods in a normal distribution. Source: This figure, inspired by a figure on page 74 of Ward and Murray (1999) [108], was drawn and has been released into the public domain by Heds 1 at English Wikipedia (Heds 1 grants anyone the right to use this work for any purpose) [109]. This figure was also slightly decorated by the author.

The Z-scores are calculated by subtracting the mean population ($\mu$ or $\bar{x}$ ) from each raw score (**X**) and then dividing the difference by the deviation from the population ($\sigma$). The process of converting raw scores into standardized scores is called normalization; however, the normalization can refer to a wide range of ratios. Calculating the Z-scores requires knowing the mean value and standard deviation of the full population (variables) to which the data points belong. If one only has a sample of population observations, then approximation with the sample mean and standard deviation gives the T-statistic. According to Kreyszig (1979) [110], if the population mean and population standard deviation are known, a raw score **X** is converted into a standard score by:

$$Z = \frac{X - \mu}{\sigma}$$

where: $\mu$ is the mean of the population. $\sigma$ is the standard deviation of the population.

The absolute value of **Z** represents the distance between the raw score **X** and the population mean in units of standard deviation. **Z** is negative when the raw score is below the average and positive when it is above the average. The standard normal distribution tables (see Appendix D) provide the "probability value" that Z, the standard normal variable, is less than a certain value. Z-values (values in the left column and on the top row) are points on horizontal scale whereas probabilities (values in the body of the table) are the regions bounded by the normal curve and horizontal scale [111].

After obtaining the Z-scores, ranking the comparative candidates (cities) were then made based on the probability values (P-values). As comparative candidates in this study are five cities, the results of ranking were accordingly ranked from one to five. Then, the scoring was made based on the ranking, which means that the first rank got five scores. The second rank got four scores. The third rank got three scores. The fourth rank got two scores. The fifth rank got one score. After obtaining the scores of the five candidates by each assessment indicator, this study then calculated the weighted scores (w-scores) by multiplying the obtained scores with the priority weights of the assessment indicators.

## 4. Results

### 4.1. Environmental Dimension

The results of the comparative assessment based on the standard scores show that the ratio of households linked to solid waste collection services in Phnom Penh is higher than in the other cities, followed by Sihanoukville and Battambang. The ratio of households installed proper toilets or adequate liquid waste storage in Battambang is much better than the other four cities, followed by Poi Pet and Sihanoukville. The public transport sharing rate of Siem Reap is greater than the other four cities whereas Sihanoukville and Phnom Penh are the second and third, respectively (see Table 10).

**Table 10.** Results of the comparative assessment for environmental indicators.

| Indicator | City | Z-Score | *p*-Value | Rank | Score |
|---|---|---|---|---|---|
| Solid waste (+) | Phnom Penh | 1.228 | 0.891 | 1 | 5 |
| | Siem Reap | −0.567 | 0.284 | 4 | 2 |
| | Battambang | −0.162 | 0.436 | 3 | 3 |
| | Poi Pet | −1.254 | 0.106 | 5 | 1 |
| | Sihanoukville | 0.756 | 0.776 | 2 | 4 |
| Wastewater (+) | Phnom Penh | −0.198 | 0.421 | 4 | 2 |
| | Siem Reap | −1.580 | 0.057 | 5 | 1 |
| | Battambang | 0.988 | 0.839 | 1 | 5 |
| | Poi Pet | 0.691 | 0.755 | 2 | 4 |
| | Sihanoukville | 0.099 | 0.536 | 3 | 3 |
| Public transport (+) | Phnom Penh | −0.987 | 0.161 | 3 | 3 |
| | Siem Reap | 1.690 | 0.955 | 1 | 5 |
| | Battambang | −0.233 | 0.109 | 4 | 2 |
| | Poi Pet | −0.274 | 0.102 | 5 | 1 |
| | Sihanoukville | −0.196 | 0.274 | 2 | 4 |
| Renewable energy (+) | Phnom Penh | −0.600 | 0.274 | 5 | 1 |
| | Siem Reap | 1.779 | 0.963 | 1 | 5 |
| | Battambang | −0.300 | 0.382 | 2 | 4 |
| | Poi Pet | −0.446 | 0.326 | 4 | 2 |
| | Sihanoukville | −0.433 | 0.334 | 3 | 3 |
| Fine dust level (–) | Phnom Penh | −0.332 | 0.371 | 2 | 4 |
| | Siem Reap | 0.535 | 0.705 | 4 | 2 |
| | Battambang | 1.221 | 0.889 | 5 | 1 |
| | Poi Pet | 0.029 | 0.618 | 3 | 3 |
| | Sihanoukville | −1.453 | 0.074 | 1 | 5 |

The ratio of households using renewable energy, solar PV, in Siem Reap is more than other cities, followed by Battambang and Sihanoukville. Finally, air quality in Sihanoukville is greater than the other four cities whereas Poi Pet and Phnom Penh are the second and third, respectively. According to the above table, Poi Pet is the lowest city in the ratio of households linked to solid waste collection services and public transport sharing rate while Siem Reap is the lowest in the ratio of households installed proper toilets/liquid waste storage. Phnom Penh is the lowest in the ratio of households using renewable energy while Battambang is the highest in PM2.5 level.

### 4.2. Social Dimension

The comparative assessment results at the social dimension of sustainability show that the ratio of households living in poor quality housings or settlement in Battambang is less than in the other cities, followed by Phnom Penh and Sihanoukville. The ratio of crimes per 10,000 households in Sihanoukville is very high compared to the other four cities, whereas Siem Reap and Phnom Penh are

the second and third ranks, respectively. The ratio of households accessing to potable water supply in Siem Reap is greater, followed by Phnom Penh and Sihanoukville (see Table 11).

**Table 11.** Results of the comparative assessment for social indicators.

| Indicator | City | Z-Score | *p*-Value | Rank | Score |
|---|---|---|---|---|---|
| | Phnom Penh | −0.473 | 0.319 | 2 | 4 |
| | Siem Reap | 1.771 | 0.962 | 5 | 1 |
| Poor housing (−) | Battambang | −0.633 | 0.264 | 1 | 5 |
| | Poi Pet | −0.232 | 0.409 | 4 | 2 |
| | Sihanoukville | −0.433 | 0.334 | 3 | 3 |
| | Phnom Penh | −0.218 | 0.413 | 4 | 2 |
| | Siem Reap | −0.413 | 0.341 | 3 | 3 |
| Crime rate (−) | Battambang | −0.569 | 0.284 | 1 | 5 |
| | Poi Pet | −0.569 | 0.284 | 1 | 5 |
| | Sihanoukville | 1.770 | 0.962 | 5 | 1 |
| | Phnom Penh | 0.699 | 0.758 | 2 | 4 |
| | Siem Reap | 0.804 | 0.788 | 1 | 5 |
| Potable water supply (+) | Battambang | −0.936 | 0.171 | 4 | 2 |
| | Poi Pet | −1.239 | 0.107 | 5 | 1 |
| | Sihanoukville | 0.672 | 0.749 | 3 | 3 |
| | Phnom Penh | 0.880 | 0.811 | 2 | 4 |
| | Siem Reap | −0.175 | 0.429 | 4 | 2 |
| Water consumption (+) | Battambang | −0.142 | 0.444 | 3 | 3 |
| | Poi Pet | −1.510 | 0.066 | 5 | 1 |
| | Sihanoukville | 0.946 | 0.829 | 1 | 5 |
| | Phnom Penh | 1.725 | 0.958 | 5 | 1 |
| | Siem Reap | −0.558 | 0.288 | 2 | 4 |
| Disaster-affected ratio (−) | Battambang | −0.051 | 0.480 | 4 | 2 |
| | Poi Pet | −0.347 | 0.363 | 3 | 3 |
| | Sihanoukville | −0.769 | 0.221 | 1 | 5 |

The ratio of households with clean-water consuming habits in Sihanoukville is better than the other cities whereas Phnom Penh and Battambang are the second and third, respectively. Finally, the ratio of households affected by disasters (windstorms and floods), per 1000 households, in Phnom Penh is more serious than the other four cities while Poi Pet and Battambang are the second and third, respectively. According to the above table, Siem Reap has the ratio of households living in poor quality houses more than other four cities. Battambang and Poi Pet are the great cities without crimes while Sihanoukville is less in the disaster-affected ratio. Poi Pet are inadequate in both potable water supply and clean-water-consuming habits, as seen lowest in the ratio of households accessing potable water supply and the ratio of households with clean-water consuming habits.

*4.3. Economic Dimension*

At the economic dimension, the comparative assessment results show that the unemployment rate in Sihanoukville is very less compared to other cities, followed by Siem Reap and Battambang. The population growth rate in Phnom Penh is greater than other cities whereas Battambang and Sihanoukville are the second and third, respectively. The labor force participation rate in Phnom Penh is very high compared to other cities, followed by Sihanoukville and Poi Pet (see Table 12).

**Table 12.** Results of the comparative assessment for economic indicators.

| Indicator | City | Z-Score | *p*-Value | Rank | Score |
|---|---|---|---|---|---|
| | Phnom Penh | 1.018 | 0.846 | 5 | 1 |
| | Siem Reap | −0.548 | 0.291 | 2 | 4 |
| Unemployment rate (−) | Battambang | 0.627 | 0.736 | 4 | 2 |
| | Poi Pet | 0.365 | 0.644 | 3 | 3 |
| | Sihanoukville | −1.462 | 0.072 | 1 | 5 |
| | Phnom Penh | 1.450 | 0.927 | 1 | 5 |
| | Siem Reap | −0.904 | 0.184 | 4 | 2 |
| Population growth (+) | Battambang | 0.329 | 0.629 | 2 | 4 |
| | Poi Pet | −0.981 | 0.164 | 5 | 1 |
| | Sihanoukville | 0.106 | 0.544 | 3 | 3 |
| | Phnom Penh | 1.014 | 0.844 | 1 | 5 |
| | Siem Reap | −0.997 | 0.161 | 4 | 2 |
| Labor participation (+) | Battambang | −1.152 | 0.125 | 5 | 1 |
| | Poi Pet | 0.511 | 0.695 | 3 | 3 |
| | Sihanoukville | 0.625 | 0.736 | 2 | 4 |
| | Phnom Penh | −0.441 | 0.330 | 4 | 1 |
| | Siem Reap | −0.433 | 0.337 | 2 | 2 |
| Near water reservoirs (+) | Battambang | −0.479 | 0.316 | 5 | 4 |
| | Poi Pet | 1.789 | 0.963 | 1 | 5 |
| | Sihanoukville | −0.436 | 0.334 | 3 | 3 |
| | Phnom Penh | 0.989 | 0.839 | 2 | 4 |
| | Siem Reap | −1.117 | 0.131 | 5 | 1 |
| Energy consumption (+) | Battambang | −0.751 | 0.227 | 4 | 2 |
| | Poi Pet | −0.202 | 0.413 | 3 | 3 |
| | Sihanoukville | 1.081 | 0.860 | 1 | 5 |

The ratio of households accessible to water reservoirs less than 150 meters in Poi Pet is greater than in the other cities, followed by Siem Reap and Sihanoukville. Finally, the ratio of households accessing the electricity supply in Sihanoukville is much better than the other four cities while Phnom Penh and Poi Pet are the second and third ranks, respectively. According to the above table, Phnom Penh has a higher unemployment rate than in the other cities while Poi Pet is very low in population growth. Battambang is very limited, compared to other cities, in both the labor force participation rate and the ratio of households accessible to water reservoirs within 150 meters. Finally, Siem Reap is lower in the ratio of households accessing to electricity supply.

*4.4. Weighting Scores of Indicators*

The weighted scores (w-scores) of indicators were calculated by multiplying the standard scores with the priority weights of indicators. The calculated w-scores of all urban sustainability indicators by each city, including their ranks are shown in Table 13. Sihanoukville was found to be strong in the environmental dimension, followed by Battambang and Phnom Penh. Furthermore, Battambang was found to be strong in social dimension whereas Sihanoukville and Phnom Penh are at the second and third ranks, respectively. Moreover, Sihanoukville was also found to be strong in the economic dimension, followed by Phnom Penh and Battambang (see Figure 9).

**Table 13.** Weighted scores of the capital and emerging secondary cities of Cambodia.

| Indicator | Weight | Phnom Penh | | Siem Reap | | Battambang | | Poi Pet | | Sihanoukville | | Standard |
|---|---|---|---|---|---|---|---|---|---|---|---|---|
| | | W-Score | Rank | W-Score | Rank | W-Score | Rank | W-Score | Rank | W-Score | Rank | |
| Solid waste | 0.0737 | 0.3685 | 1 | 0.1474 | 4 | 0.2211 | 3 | 0.0737 | 5 | 0.2948 | 2 | 0.2211 |
| Wastewater | 0.0652 | 0.1304 | 4 | 0.0652 | 5 | 0.3260 | 1 | 0.2608 | 2 | 0.1956 | 3 | 0.1956 |
| Public transport | 0.0568 | 0.1704 | 3 | 0.2840 | 1 | 0.1136 | 4 | 0.0568 | 5 | 0.2272 | 2 | 0.1704 |
| Renewable energy | 0.0478 | 0.0478 | 5 | 0.2390 | 1 | 0.1912 | 2 | 0.0956 | 4 | 0.1434 | 3 | 0.1434 |
| Fine dust level | 0.0424 | 0.1696 | 2 | 0.0848 | 4 | 0.0424 | 5 | 0.1272 | 3 | 0.2120 | 1 | 0.1272 |
| **Environmental** | **-** | **0.8867** | **3** | **0.8204** | **4** | **0.8943** | **2** | **0.6141** | **5** | **1.0730** | **1** | **0.8577** |
| Poor housing | 0.0857 | 0.3428 | 2 | 0.0857 | 5 | 0.4285 | 1 | 0.1714 | 4 | 0.2571 | 3 | 0.2571 |
| Crime rate | 0.0770 | 0.1540 | 4 | 0.2310 | 3 | 0.3850 | 1 | 0.3850 | 1 | 0.0770 | 5 | 0.2464 |
| Potable water supply | 0.0769 | 0.3076 | 2 | 0.3845 | 1 | 0.1538 | 4 | 0.0769 | 5 | 0.2307 | 3 | 0.2307 |
| Water consumption | 0.0620 | 0.2480 | 2 | 0.1240 | 4 | 0.1860 | 3 | 0.0620 | 5 | 0.3100 | 1 | 0.1860 |
| Disaster-affected ratio | 0.0574 | 0.0574 | 5 | 0.2296 | 2 | 0.1148 | 4 | 0.1722 | 3 | 0.2870 | 1 | 0.1722 |
| **Social** | **-** | **1.1098** | **3** | **1.0548** | **4** | **1.2681** | **1** | **0.8675** | **5** | **1.1618** | **2** | **1.0924** |
| Unemployment rate | 0.0816 | 0.0816 | 5 | 0.3264 | 2 | 0.1632 | 4 | 0.2448 | 3 | 0.4080 | 1 | 0.2448 |
| Population growth | 0.0762 | 0.3810 | 1 | 0.1524 | 4 | 0.3048 | 2 | 0.0762 | 5 | 0.2286 | 3 | 0.2286 |
| Labor participation | 0.0721 | 0.3605 | 1 | 0.1442 | 4 | 0.0721 | 5 | 0.2163 | 3 | 0.2884 | 2 | 0.2163 |
| Near water reservoir | 0.0662 | 0.0662 | 5 | 0.1324 | 4 | 0.2648 | 2 | 0.3310 | 1 | 0.1986 | 3 | 0.1986 |
| Energy consumption | 0.0584 | 0.2336 | 2 | 0.0584 | 5 | 0.1168 | 4 | 0.1752 | 3 | 0.2920 | 1 | 0.1752 |
| **Economic** | **-** | **1.1229** | **2** | **0.8138** | **5** | **0.9217** | **4** | **1.0435** | **3** | **1.4156** | **1** | **1.0635** |
| **Sustainability** | **1.0000** | **3.1194** | **2** | **2.6890** | **4** | **3.0841** | **3** | **2.5251** | **5** | **3.6504** | **1** | **3.0136** |

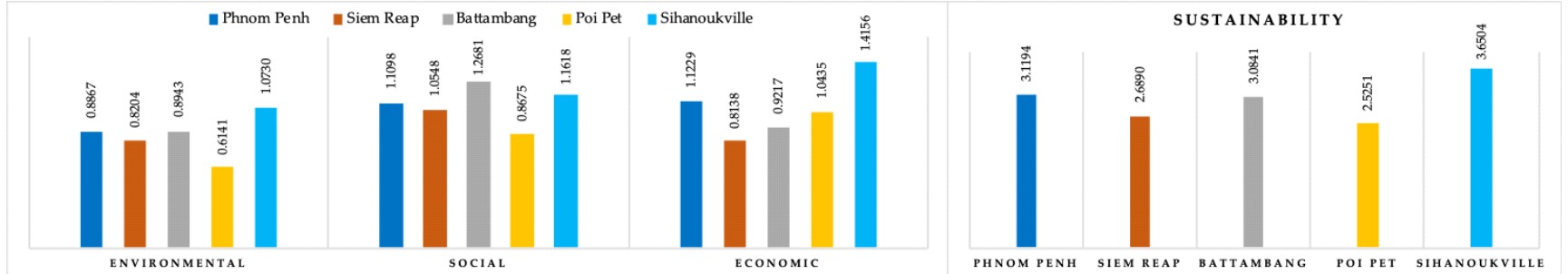

**Figure 9.** Sustainability level of the capital and emerging secondary cities of Cambodia.

Poi Pet was found to be the weakest in both environmental and social dimensions, compared to the other four cities whereas Siem Reap was found to be the weakest in the economic dimension. Totally, the sustainability level of Sihanoukville is the strongest, followed by Phnom Penh and Battambang. Sihanoukville is very strong in the sustainability level because this city has significantly achieved five indicators with the first rank. Those indicators are air quality (good), clean-water-consuming habits (more), disaster-affected ratio (low), unemployment rate (less), and electricity supply (high).

*4.5. Standard of Urban Sustainability in Cambodia*

Based on the standard scores and priority weights of the indicators, the standard for assessing the sustainability of the capital and emerging secondary cities of Cambodia was developed as shown in Table 14. Based on this standard, the housing indicator is given high priority, followed by employment and crime indicators, whereas the lowest priority is fine dust level (see Figure 10). Consequently, this standard will be used to discuss the strengths and weaknesses in the sustainability level of the capital and emerging secondary cities of Cambodia by each indicator.

**Table 14.** Z-score- and priority-based standard for measuring the sustainability of five cities.

| Dimension | Indicator | Standard |
|---|---|---|
| Environmental | Solid waste | 0.2211 |
| | Wastewater | 0.1956 |
| | Public transport | 0.1704 |
| | Renewable energy | 0.1434 |
| | Fine dust level | 0.1272 |
| Social | Poor housing | 0.2571 |
| | Crime rate | 0.2464 |
| | Potable water supply | 0.2307 |
| | Water consumption | 0.1860 |
| | Disaster-affected ratio | 0.1722 |
| Economic | Unemployment rate | 0.2448 |
| | Population growth rate | 0.2286 |
| | Labor participation | 0.2163 |
| | Near water reservoir | 0.1986 |
| | Energy consumption | 0.1752 |

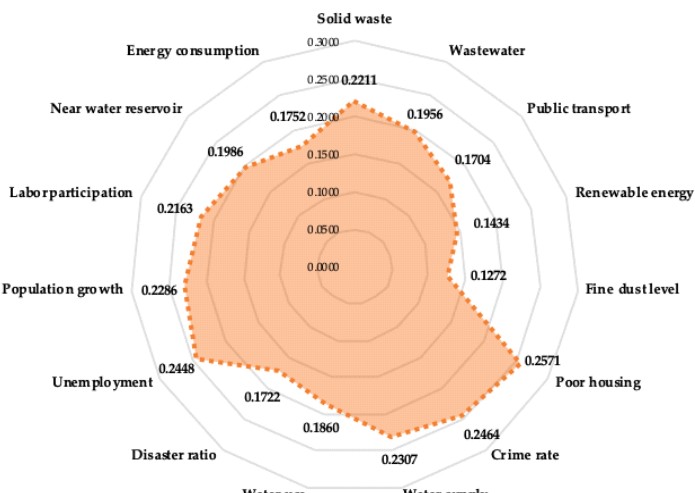

**Figure 10.** Standard for measuring the sustainability of five cities in Cambodia.

## 5. Discussion and Implications

### 5.1. Sustainability of Phnom Penh

Phnom Penh has achieved nine indicators above the standard whereas six indicators were found to be lower than the standard. As shown in Figure 11, the sustainability level of Phnom Penh is high in solid waste collection services, public transport sharing rate, air quality, quality housing, potable water supply infrastructure, clean-water-consuming behaviors, population growth rate, labor force participation rate, and electricity supplying rate. However, the sustainability level of this capital is low in proper toilets/wastewater storages, renewable energy consumption and promotion, crime prevention, disaster risk management, employment, and accessible water reservoirs. Thus, in order to improve its sustainability level, this capital should further consider expanding proper public toilets and wastewater management systems, promoting renewable energy consumption and production, preventing crimes, expanding disaster prevention facilities, reducing the unemployment rate by creating new jobs, and preserving or constructing water reservoirs.

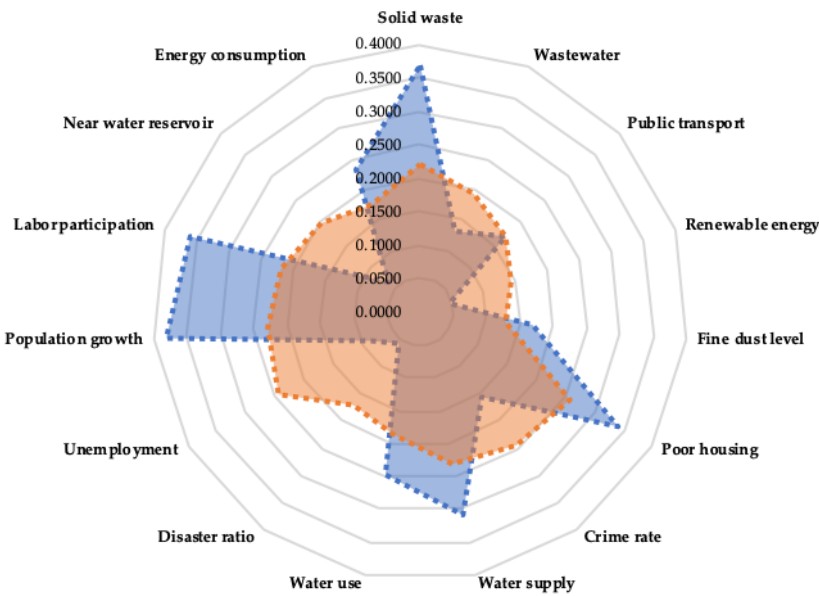

**Figure 11.** Sustainability level of Phnom Penh by indicator.

### 5.2. Sustainability of Siem Reap

Siem Reap has achieved five indicators above the standard whereas ten indicators were found to be lower than the standard. As shown in Figure 12, the sustainability level of Siem Reap is high in public transport sharing rate, renewable energy consumption, potable water supply infrastructure, disaster risk management, and employment. However, the sustainability level of this city is low in solid waste collection services, proper toilets and wastewater storages, air quality, poor (informal or unplanned) settlements, crime prevention, clean-water-consuming behaviors, population growth rate, labor force participation rate, accessible water reservoirs, and electricity supplying rate. Therefore, in order to improve its sustainability level, this city should further consider improving solid waste collection services, expanding proper public toilets and wastewater storages, improving air quality by promoting environmentally friendly transport means, upgrading poor housing and prohibiting informal and unplanned settlements, preventing crimes, rising awareness on clean water consumption, creating new and stable jobs for urban residents beyond current tourism-based jobs towards reducing the migration rate of the residents to work in other cities or countries, preserving or constructing water reservoirs, and expanding electricity supply in the city.

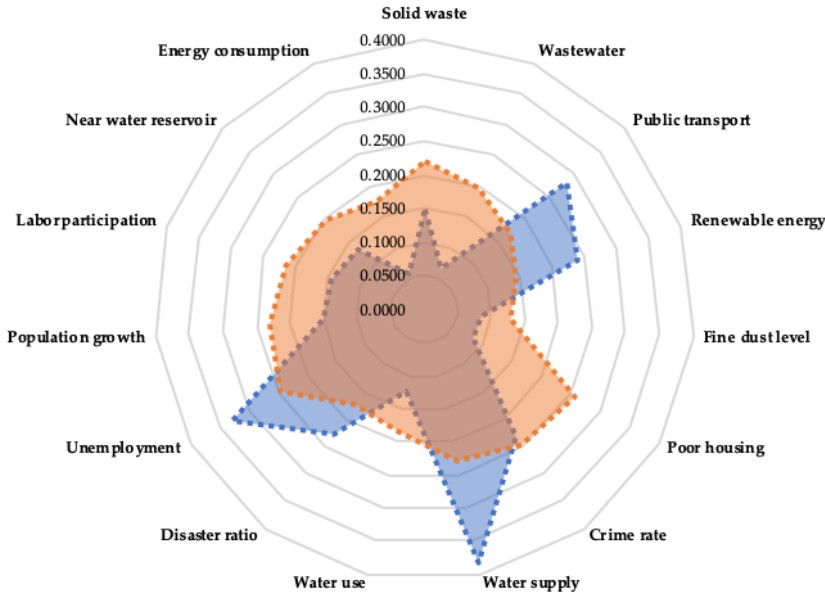

**Figure 12.** Sustainability level of Siem Reap by indicator.

### 5.3. Sustainability of Battambang

Battambang has achieved eight indicators above the standard whereas seven indicators were found to be lower than the standard. As shown in Figure 13, the sustainability level of Battambang is high in solid waste collection services, proper toilets and wastewater storages, renewable energy consumption, quality housings and settlements, crime prevention, clean-water-consuming behaviors, population growth rate, and accessible water reservoirs. However, the sustainability level of this city is low in public transport sharing rate, air quality, potable water supply ratio, disaster-affected ratio, unemployment rate, labor force participation rate, and electricity supply rate. Consequently, in order to improve its sustainability level, this city should further consider expanding public transport sharing rate, improving air quality by promoting environmentally friendly transport means, increasing potable water supply, expanding disaster prevention facilities, creating new and stable jobs for residents towards reducing the unemployment rate and migration rate of the residents to work in other cities or countries, and increasing electricity supply.

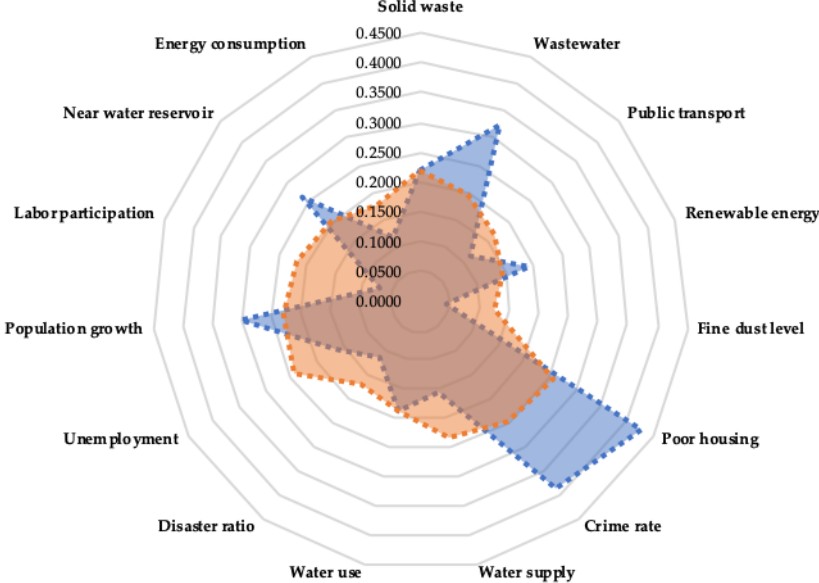

**Figure 13.** Sustainability level of Battambang by indicator.

### 5.4. Sustainability of Poi Pet

Poi Pet has achieved eight indicators above the standard whereas seven indicators were found to be lower than the standard. As shown in Figure 14, the sustainability level of Poi Pet is high in proper toilets and wastewater storages, air quality, crime prevention, disaster risk prevention, employment rate, labor force participation rate, accessible water reservoirs, and electricity supply rate. However, the sustainability level of this city is low in solid waste collection services, public transport sharing rate, renewable energy consumption and production, quality housings and settlements, potable water supply ratio, clean-water-consuming behaviors, and population growth rate. Therefore, in order to improve its sustainability level, this city should further consider improving solid waste collection services, promoting public transport means in the city, encouraging renewable energy consumption and production, upgrading poor housing and prohibiting informal/unplanned settlements, increasing potable water supplying ratio, rising awareness on clean-water-consuming behaviors, and creating new and stable jobs for residents towards reducing the migration rate to working in other cities or countries.

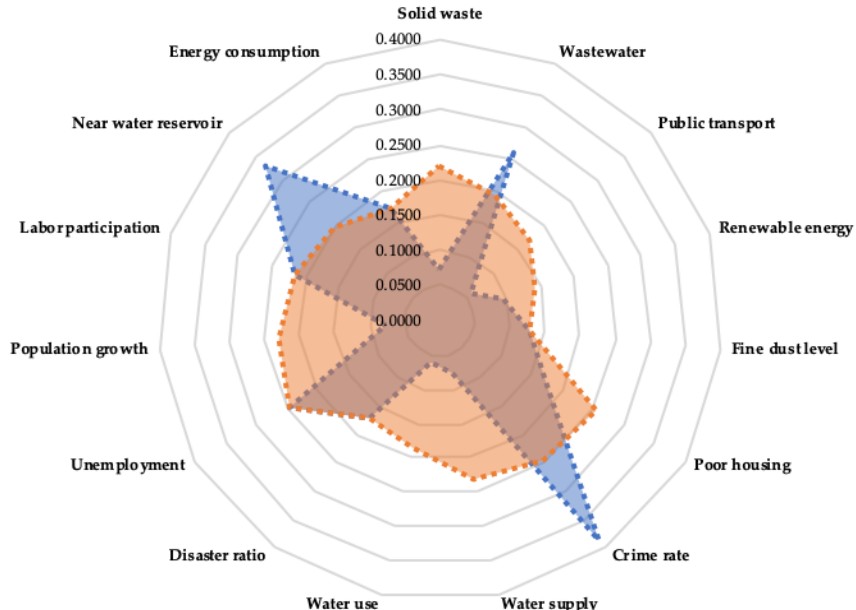

**Figure 14.** Sustainability level of Poi Pet by indicator.

### 5.5. Sustainability of Sihanoukville

Sihanoukville has significantly achieved 14 indicators above the standard whereas only one indicator is lower than the standard. That indicator is the crime rate (see Figure 15). Consequently, the sustainability level of this city was found to be great in environmental and economic dimensions while the social dimension is at the second rank after Battambang. This attribute makes Sihanoukville stronger than the other four cities in the sustainability level. As this city is growing very fast, it is quite challenging in terms of security at nighttime; that is why its crime rate is very high, compared to the other four cities. Therefore, in order to improve its sustainability level in terms of social safety and security, this city should focus on improving safety systems in the city, initiating crime prevention measures, and/or increasing patrol police officers at night.

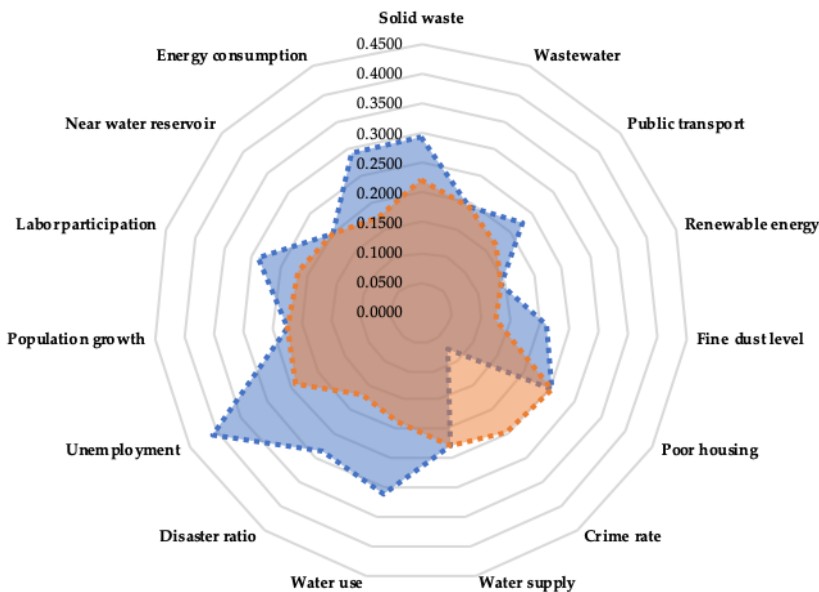

**Figure 15.** Sustainability level of Sihanoukville by indicator.

*5.6. Significance of the Priority Weights of Indicators in Urban Sustainability Assessment*

The total unweighted scores for each city by sustainability dimension resulted from the summed-up standard scores as shown in Table 10 (environmental sustainability), Table 11 (social sustainability), and Table 12 (economic sustainability). These total unweighted scores, as well as the total weighted scores by sustainability dimension (Table 13), are shown, respectively, with their ranks in Table 15 below. According to this table, the ranks of two cities for each of the environmental and social sustainability dimensions based on the total unweighted scores are different from their ranks based on the total weighted scores. Therefore, the comparative assessment of cities is required using the priority weights of indicators in order to improve the accuracy of the comparison.

**Table 15.** Ranks of the five cities based on unweighted and weighted scores of indicators.

| Sustainability Dimension | City | Unweighted | | Weighted | |
|---|---|---|---|---|---|
| | | Total Score | Rank | Total Score | Rank |
| Environmental | Phnom Penh | 15 | **2** | 0.8866 | **3** |
| | Siem Reap | 15 | **2** | 0.8204 | **4** |
| | Battambang | 15 | 2 | 0.8943 | 2 |
| | Poi Pet | 11 | 5 | 0.6141 | 5 |
| | Sihanoukville | 19 | 1 | 1.0730 | 1 |
| Social | Phnom Penh | 15 | 3 | 1.1098 | 3 |
| | Siem Reap | 15 | 3 | 1.0548 | 4 |
| | Battambang | 17 | 1 | 1.2681 | 1 |
| | Poi Pet | 12 | 5 | 0.8675 | 5 |
| | Sihanoukville | 17 | **1** | 1.1618 | **2** |
| Economic | Phnom Penh | 16 | 2 | 1.1229 | 2 |
| | Siem Reap | 11 | 5 | 0.8138 | 5 |
| | Battambang | 13 | 4 | 0.9217 | 4 |
| | Poi Pet | 15 | 3 | 1.0435 | 3 |
| | Sihanoukville | 20 | 1 | 1.4156 | 1 |

## 6. Conclusions

The sustainability level of Sihanoukville is significantly strong, compared to the other four cities. The levels of environmental, social, and economic sustainability are all above the average. Particularly,

the levels of environmental and economic sustainability are at the first rank whereas social sustainability is at the second rank after Battambang.

Phnom Penh is also very strong in the sustainability level. The total sustainability level is at the second rank after Sihanoukville. The levels of environmental, social, and economic sustainability are all above the average. The levels of environmental and social sustainability are at the third rank whereas economic sustainability is at the second rank after Sihanoukville.

The sustainability level of Battambang is moderately strong. The total sustainability level is above the average although economic sustainability is lower than the average. Particularly, the levels of environmental and social sustainability are all above average. The level of social sustainability is at the first rank whereas environmental sustainability is at the second rank.

Siem Reap is at the fourth rank. The levels of environmental and social sustainability are slightly lower than the average. However, the level of economic sustainability is not much lower than the average. The sustainability level of Poi Pet is significantly low compared to the other four cities. Even though the level of economic sustainability is slightly lower than the average, the levels of environmental and social sustainability are remarkably low.

The ranks of sustainability levels of the five cities based on weighted scores are different from their ranks based on unweighted scores. Therefore, priority weights of indicators are necessary for the comparative assessment towards improving the accuracy of the comparison.

The following could be the limitation of this study. The data used were based on the 2018 Commune Database published in early 2019 by the Capital and Provincial Departments of Planning. In this database, motorcycles were not separated between private and public (moto-dub). The total inland transport means also excluded motorcycles. Therefore, the calculation of the public transport sharing rate in this study did not include public motorcycles.

**Funding:** This research received no external funding.

**Acknowledgments:** The author expresses his profound gratitude to the Department of Green Economy, General Secretariat of the National Council for Sustainable Development, and the Ministry of Environment (Cambodia) who administratively supported this research, provided the place for Delphi panel surveys, and helped invite the panelists and the National Institute of International Education of the Ministry of Education (South Korea) who sponsored his doctoral research since the beginning until he has successfully completed his degree. A part of this study is published in his doctoral dissertation at the Hanyang Paiknam Library.

**Conflicts of Interest:** The author declares no conflict of interest.

## Appendix A. Urban Sustainability Index

**Table A1.** Conversion of 100-scaled urban sustainability index for Cambodia.

| Indicator | Dimension | Weight | | | |
|---|---|---|---|---|---|
| | | Origin | Conversion | 100-Scale | In Percentage |
| **Solid waste collection rate** | | 0.0437 | 0.0437+(0.4506÷15) | 0.0737 | 7.37% |
| Wastewater treatment ratio | | 0.0352 | 0.0352+(0.4506÷15) | 0.0652 | 6.52% |
| Public transport sharing rate | Environmental | 0.0268 | 0.0268+(0.4506÷15) | 0.0568 | 5.68% |
| Renewable energy shared rate | | 0.0178 | 0.0178+(0.4506÷15) | 0.0478 | 4.78% |
| Fine dust level (PM2.5, PM10) | | 0.0124 | 0.0124+(0.4506÷15) | 0.0424 | 4.24% |
| Slum/poor settlement rate | | 0.0557 | 0.0557+(0.4506÷15) | 0.0857 | 8.57% |
| Crime rate or prevention | | 0.0470 | 0.0470+(0.4506÷15) | 0.0770 | 7.70% |
| Potable water supply ratio | Social | 0.0469 | 0.0469+(0.4506÷15) | 0.0769 | 7.69% |
| Water consumption rate | | 0.0320 | 0.0320+(0.4506÷15) | 0.0620 | 6.20% |
| Disaster prevention | | 0.0274 | 0.0274+(0.4506÷15) | 0.0574 | 5.74% |
| Unemployment rate | | 0.0516 | 0.0516+(0.4506÷15) | 0.0816 | 8.16% |
| Population growth rate | | 0.0462 | 0.0462+(0.4506÷15) | 0.0762 | 7.62% |
| Labor force participation rate | Economic | 0.0421 | 0.0421+(0.4506÷15) | 0.0721 | 7.21% |
| Water reservoir coverage | | 0.0362 | 0.0362+(0.4506÷15) | 0.0662 | 6.62% |
| Energy consumption rate | | 0.0284 | 0.0284+(0.4506÷15) | 0.0584 | 5.84% |
| Total | Sustainability | 0.5494 | 0.5494 + 0.4506 | 1.0000 | 100.00% |

## Appendix B. Sources of Data

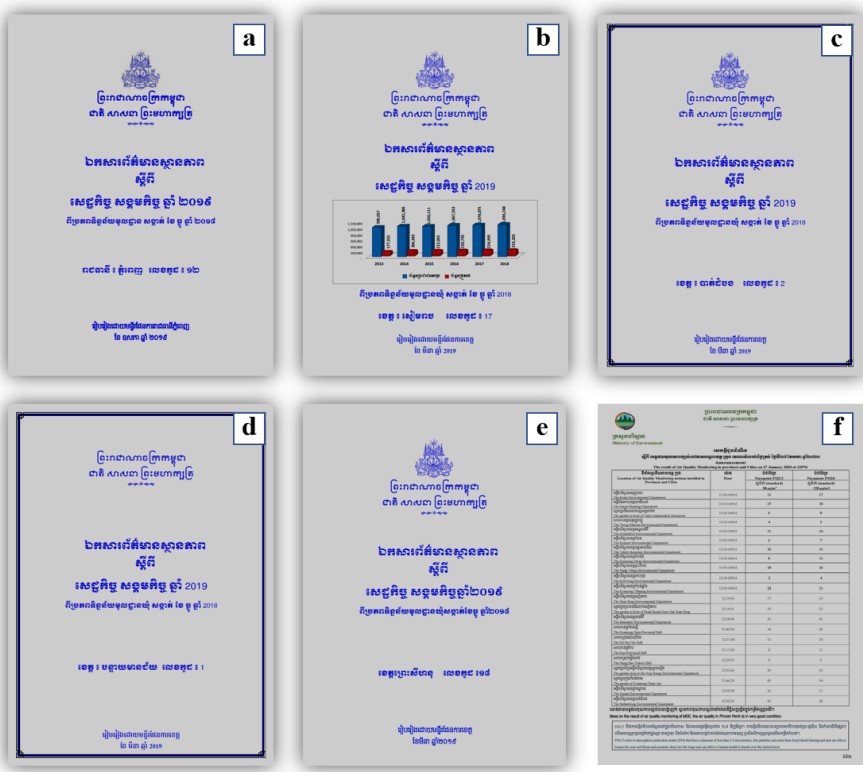

**Figure A1.** Sources of data. (**a**) was published by the Phnom Penh Capital Department of Planning (DoP). (**b**) was published by the Siem Reap Provincial DoP. (**c**) was published by the Battambang Provincial DoP. (**d**) was published by the Banteay Meanchey Provincial DoP. (**e**) was published by Preah Sihanouk Provincial DoP. (**f**) was published by the Ministry of Environment.

## Appendix C. Calculation of Data

**Table A2.** Calculation of public transport sharing rate based on the 2018 commune database [59–61,103,104].

| Transport Mean | Phnom Penh | Siem Reap | Battambang | Poi Pet | Sihanoukville |
|---|---|---|---|---|---|
| Total inland transport means | 101,080 | 20,000 | 17,819 | 13,606 | 8946 |
| Small passenger cars | 8647 | 2737 | 1467 | 1242 | 463 |
| Large passenger cars | 1762 | 1263 | 843 | 544 | 638 |
| Tricycles | 641 | 507 | 221 | 121 | 184 |
| Total public transport means | 11,050 | 4507 | 2531 | 1907 | 1285 |
| **Public transport sharing rate** | **10.93** | **22.54** | **14.20** | **14.02** | **14.36** |

**Table A3.** Calculation of the average level of fine dust (PM2.5), measured by the Ministry of Environment.

| Date | Phnom Penh | Siem Reap | Battambang | Poi Pet | Sihanoukville | Source |
|---|---|---|---|---|---|---|
| 27 January 2020 | 29 | 17 | 20 | 13 | 8 | [105] |
| 31 January 2020 | 30 | 48 | 65 | 44 | 20 | [106] |
| 18 February 2020 | 12 | 30 | 29 | 24 | 12 | [107] |
| **Average** | **24** | **32** | **38** | **27** | **13** | **-** |

**Table A4.** Calculation of the ratio of households affected by disasters based on the 2018 Commune Database.

| Type of Disaster | Phnom Penh | Siem Reap | Battambang | Poi Pet | Sihanoukville |
|:---:|:---:|:---:|:---:|:---:|:---:|
| Windstorm | 0.1 | 0.3 | 0.2 | 0.2 | 0.1 |
| Floods | 5.9 | 0.3 | 1.6 | 0.9 | 0.0 |
| **Total** | 6.0 | 0.6 | 1.8 | 1.1 | 0.1 |

**Table A5.** Calculation of labor force participation and unemployment rate based on the 2018 commune database.

| Category | Phnom Penh | Siem Reap | Battambang | Poi Pet | Sihanoukville |
|:---|:---:|:---:|:---:|:---:|:---:|
| Total provincial populations | 1,474,489 | 1,096,248 | 1,312,051 | 789,953 | 221,360 |
| Populations in labor force | 862,097 | 576,946 | 684,604 | 450,387 | 126,938 |
| Labor force participation rate | 58.47 | 52.63 | 52.18 | 57.01 | 57.34 |
| Total employed populations | 538,058 | 493,881 | 467,033 | 323,617 | 127,262 |
| Unemployed populations | 324,039 | 83,065 | 217,571 | 126,770 | −324 |
| Unemployment rate | 0.38 | 0.14 | 0.32 | 0.28 | 0.00 |

## Appendix D. Standard Z-Score Table

**Table A6.** Z-score table.

| | Negative Standard Score | | | | | | | | | | | Positive Standard Score | | | | | | | | | |
|---|---|---|---|---|---|---|---|---|---|---|---|---|---|---|---|---|---|---|---|---|---|
| Z | 0.00 | 0.01 | 0.02 | 0.03 | 0.04 | 0.05 | 0.06 | 0.07 | 0.08 | 0.09 | Z | 0.00 | 0.01 | 0.02 | 0.03 | 0.04 | 0.05 | 0.06 | 0.07 | 0.08 | 0.09 |
| −3.5 | 0.0002 | 0.0003 | 0.0003 | 0.0003 | 0.0003 | 0.0003 | 0.0003 | 0.0003 | 0.0003 | 0.0003 | 0.0 | 0.5000 | 0.5040 | 0.5080 | 0.5120 | 0.5160 | 0.5199 | 0.5239 | 0.5279 | 0.5319 | 0.5359 |
| −3.4 | 0.0003 | 0.0003 | 0.0004 | 0.0004 | 0.0004 | 0.0004 | 0.0004 | 0.0004 | 0.0005 | 0.0005 | 0.1 | 0.5398 | 0.5438 | 0.5478 | 0.5517 | 0.5557 | 0.5596 | 0.5636 | 0.5675 | 0.5714 | 0.5753 |
| −3.3 | 0.0005 | 0.0005 | 0.0005 | 0.0005 | 0.0006 | 0.0006 | 0.0006 | 0.0006 | 0.0006 | 0.0007 | 0.2 | 0.5793 | 0.5832 | 0.5871 | 0.5910 | 0.5948 | 0.5987 | 0.6026 | 0.6064 | 0.6103 | 0.6141 |
| −3.2 | 0.0007 | 0.0007 | 0.0007 | 0.0008 | 0.0008 | 0.0008 | 0.0008 | 0.0009 | 0.0009 | 0.0009 | 0.3 | 0.6179 | 0.6217 | 0.6255 | 0.6293 | 0.6331 | 0.6368 | 0.6406 | 0.6443 | 0.6480 | 0.6517 |
| −3.1 | 0.0010 | 0.0010 | 0.0010 | 0.0011 | 0.0011 | 0.0011 | 0.0012 | 0.0012 | 0.0013 | 0.0013 | 0.4 | 0.6554 | 0.6591 | 0.6628 | 0.6664 | 0.6700 | 0.6736 | 0.6772 | 0.6808 | 0.6844 | 0.6879 |
| −3.0 | 0.0013 | 0.0014 | 0.0014 | 0.0015 | 0.0015 | 0.0016 | 0.0016 | 0.0017 | 0.0018 | 0.0018 | 0.5 | 0.6915 | 0.6950 | 0.6985 | 0.7019 | 0.7054 | 0.7088 | 0.7123 | 0.7157 | 0.7190 | 0.7224 |
| −2.9 | 0.0019 | 0.0019 | 0.0020 | 0.0021 | 0.0021 | 0.0022 | 0.0023 | 0.0023 | 0.0024 | 0.0025 | 0.6 | 0.7257 | 0.7291 | 0.7324 | 0.7357 | 0.7389 | 0.7422 | 0.7454 | 0.7486 | 0.7517 | 0.7549 |
| −2.8 | 0.0026 | 0.0026 | 0.0027 | 0.0028 | 0.0029 | 0.0030 | 0.0031 | 0.0032 | 0.0033 | 0.0034 | 0.7 | 0.7580 | 0.7611 | 0.7642 | 0.7673 | 0.7704 | 0.7734 | 0.7764 | 0.7794 | 0.7823 | 0.7852 |
| −2.7 | 0.0035 | 0.0036 | 0.0037 | 0.0038 | 0.0039 | 0.0040 | 0.0041 | 0.0043 | 0.0044 | 0.0045 | 0.8 | 0.7881 | 0.7910 | 0.7939 | 0.7967 | 0.7995 | 0.8023 | 0.8051 | 0.8078 | 0.8106 | 0.8133 |
| −2.6 | 0.0047 | 0.0048 | 0.0049 | 0.0051 | 0.0052 | 0.0054 | 0.0055 | 0.0057 | 0.0059 | 0.0060 | 0.9 | 0.8159 | 0.8186 | 0.8212 | 0.8238 | 0.8264 | 0.8289 | 0.8315 | 0.8340 | 0.8365 | 0.8389 |
| −2.5 | 0.0062 | 0.0064 | 0.0066 | 0.0068 | 0.0069 | 0.0071 | 0.0073 | 0.0075 | 0.0078 | 0.0080 | 1.0 | 0.8413 | 0.8438 | 0.8461 | 0.8485 | 0.8508 | 0.8531 | 0.8554 | 0.8577 | 0.8599 | 0.8621 |
| −2.4 | 0.0082 | 0.0084 | 0.0087 | 0.0089 | 0.0091 | 0.0094 | 0.0096 | 0.0099 | 0.0102 | 0.0104 | 1.1 | 0.8643 | 0.8665 | 0.8686 | 0.8708 | 0.8729 | 0.8749 | 0.8770 | 0.8790 | 0.8810 | 0.8830 |
| −2.3 | 0.0107 | 0.0110 | 0.0113 | 0.0116 | 0.0119 | 0.0122 | 0.0125 | 0.0129 | 0.0132 | 0.0136 | 1.2 | 0.8849 | 0.8869 | 0.8888 | 0.8907 | 0.8925 | 0.8944 | 0.8962 | 0.8980 | 0.8997 | 0.9015 |
| −2.2 | 0.0139 | 0.0143 | 0.0146 | 0.0150 | 0.0154 | 0.0158 | 0.0162 | 0.0166 | 0.0170 | 0.0174 | 1.3 | 0.9032 | 0.9049 | 0.9066 | 0.9082 | 0.9099 | 0.9115 | 0.9131 | 0.9147 | 0.9162 | 0.9177 |
| −2.1 | 0.0179 | 0.0183 | 0.0188 | 0.0192 | 0.0197 | 0.0202 | 0.0207 | 0.0212 | 0.0217 | 0.0222 | 1.4 | 0.9192 | 0.9207 | 0.9222 | 0.9236 | 0.9251 | 0.9265 | 0.9279 | 0.9292 | 0.9306 | 0.9319 |
| −2.0 | 0.0228 | 0.0233 | 0.0239 | 0.0244 | 0.0250 | 0.0256 | 0.0262 | 0.0268 | 0.0274 | 0.0281 | 1.5 | 0.9332 | 0.9345 | 0.9357 | 0.9370 | 0.9382 | 0.9394 | 0.9406 | 0.9418 | 0.9429 | 0.9441 |
| −1.9 | 0.0287 | 0.0294 | 0.0301 | 0.0307 | 0.0314 | 0.0322 | 0.0329 | 0.0336 | 0.0344 | 0.0351 | 1.6 | 0.9452 | 0.9463 | 0.9474 | 0.9484 | 0.9495 | 0.9505 | 0.9515 | 0.9525 | 0.9535 | 0.9545 |
| −1.8 | 0.0359 | 0.0367 | 0.0375 | 0.0384 | 0.0392 | 0.0401 | 0.0409 | 0.0418 | 0.0427 | 0.0436 | 1.7 | 0.9554 | 0.9564 | 0.9573 | 0.9582 | 0.9591 | 0.9599 | 0.9608 | 0.9616 | 0.9625 | 0.9633 |
| −1.7 | 0.0446 | 0.0455 | 0.0465 | 0.0475 | 0.0485 | 0.0495 | 0.0505 | 0.0516 | 0.0526 | 0.0537 | 1.8 | 0.9641 | 0.9649 | 0.9656 | 0.9664 | 0.9671 | 0.9678 | 0.9686 | 0.9693 | 0.9699 | 0.9706 |
| −1.6 | 0.0548 | 0.0559 | 0.0571 | 0.0582 | 0.0594 | 0.0606 | 0.0618 | 0.0630 | 0.0643 | 0.0655 | 1.9 | 0.9713 | 0.9719 | 0.9726 | 0.9732 | 0.9738 | 0.9744 | 0.9750 | 0.9756 | 0.9761 | 0.9767 |
| −1.5 | 0.0668 | 0.0681 | 0.0694 | 0.0708 | 0.0721 | 0.0735 | 0.0749 | 0.0764 | 0.0778 | 0.0793 | 2.0 | 0.9772 | 0.9778 | 0.9783 | 0.9788 | 0.9793 | 0.9798 | 0.9803 | 0.9808 | 0.9812 | 0.9817 |
| −1.4 | 0.0808 | 0.0823 | 0.0838 | 0.0853 | 0.0869 | 0.0885 | 0.0901 | 0.0918 | 0.0934 | 0.0951 | 2.1 | 0.9821 | 0.9826 | 0.9830 | 0.9834 | 0.9838 | 0.9842 | 0.9846 | 0.9850 | 0.9854 | 0.9857 |
| −1.3 | 0.0968 | 0.0985 | 0.1003 | 0.1020 | 0.1038 | 0.1056 | 0.1075 | 0.1093 | 0.1112 | 0.1131 | 2.2 | 0.9861 | 0.9864 | 0.9868 | 0.9871 | 0.9875 | 0.9878 | 0.9881 | 0.9884 | 0.9887 | 0.9890 |
| −1.2 | 0.1151 | 0.1170 | 0.1190 | 0.1210 | 0.1230 | 0.1251 | 0.1271 | 0.1292 | 0.1314 | 0.1335 | 2.3 | 0.9893 | 0.9896 | 0.9898 | 0.9901 | 0.9904 | 0.9906 | 0.9909 | 0.9911 | 0.9913 | 0.9916 |
| −1.1 | 0.1357 | 0.1379 | 0.1401 | 0.1423 | 0.1446 | 0.1469 | 0.1492 | 0.1515 | 0.1539 | 0.1562 | 2.4 | 0.9918 | 0.9920 | 0.9922 | 0.9925 | 0.9927 | 0.9929 | 0.9931 | 0.9932 | 0.9934 | 0.9936 |
| −1.0 | 0.1587 | 0.1611 | 0.1635 | 0.1660 | 0.1685 | 0.1711 | 0.1736 | 0.1762 | 0.1788 | 0.1814 | 2.5 | 0.9938 | 0.9940 | 0.9941 | 0.9943 | 0.9945 | 0.9946 | 0.9948 | 0.9949 | 0.9951 | 0.9952 |
| −0.9 | 0.1841 | 0.1867 | 0.1894 | 0.1922 | 0.1949 | 0.1977 | 0.2005 | 0.2033 | 0.2061 | 0.2090 | 2.6 | 0.9953 | 0.9955 | 0.9956 | 0.9957 | 0.9959 | 0.9960 | 0.9961 | 0.9962 | 0.9963 | 0.9964 |
| −0.8 | 0.2119 | 0.2148 | 0.2177 | 0.2206 | 0.2236 | 0.2266 | 0.2296 | 0.2327 | 0.2358 | 0.2389 | 2.7 | 0.9965 | 0.9966 | 0.9967 | 0.9968 | 0.9969 | 0.9970 | 0.9971 | 0.9972 | 0.9973 | 0.9974 |
| −0.7 | 0.2420 | 0.2451 | 0.2483 | 0.2514 | 0.2546 | 0.2578 | 0.2611 | 0.2643 | 0.2676 | 0.2709 | 2.8 | 0.9974 | 0.9975 | 0.9976 | 0.9977 | 0.9977 | 0.9978 | 0.9979 | 0.9979 | 0.9980 | 0.9981 |
| −0.6 | 0.2743 | 0.2776 | 0.2810 | 0.2843 | 0.2877 | 0.2912 | 0.2946 | 0.2981 | 0.3015 | 0.3050 | 2.9 | 0.9981 | 0.9982 | 0.9982 | 0.9983 | 0.9984 | 0.9984 | 0.9985 | 0.9985 | 0.9986 | 0.9986 |
| −0.5 | 0.3085 | 0.3121 | 0.3156 | 0.3192 | 0.3228 | 0.3264 | 0.3300 | 0.3336 | 0.3372 | 0.3409 | 3.0 | 0.9987 | 0.9987 | 0.9987 | 0.9988 | 0.9988 | 0.9989 | 0.9989 | 0.9989 | 0.9990 | 0.9990 |
| −0.4 | 0.3446 | 0.3483 | 0.3520 | 0.3557 | 0.3594 | 0.3632 | 0.3669 | 0.3707 | 0.3745 | 0.3783 | 3.1 | 0.9990 | 0.9991 | 0.9991 | 0.9991 | 0.9992 | 0.9992 | 0.9992 | 0.9992 | 0.9993 | 0.9993 |
| −0.3 | 0.3821 | 0.3859 | 0.3897 | 0.3936 | 0.3974 | 0.4013 | 0.4052 | 0.4090 | 0.4129 | 0.4168 | 3.2 | 0.9993 | 0.9993 | 0.9994 | 0.9994 | 0.9994 | 0.9994 | 0.9994 | 0.9995 | 0.9995 | 0.9995 |
| −0.2 | 0.4207 | 0.4247 | 0.4286 | 0.4325 | 0.4364 | 0.4404 | 0.4443 | 0.4483 | 0.4522 | 0.4562 | 3.3 | 0.9995 | 0.9995 | 0.9995 | 0.9996 | 0.9996 | 0.9996 | 0.9996 | 0.9996 | 0.9996 | 0.9997 |
| −0.1 | 0.4602 | 0.4641 | 0.4681 | 0.4721 | 0.4761 | 0.4801 | 0.4840 | 0.4880 | 0.4920 | 0.4960 | 3.4 | 0.9997 | 0.9997 | 0.9997 | 0.9997 | 0.9997 | 0.9997 | 0.9997 | 0.9997 | 0.9997 | 0.9998 |
| 0.0 | 0.5000 | 0.5040 | 0.5080 | 0.5120 | 0.5160 | 0.5199 | 0.5239 | 0.5279 | 0.5319 | 0.5359 | 3.5 | 0.9998 | 0.9998 | 0.9998 | 0.9998 | 0.9998 | 0.9998 | 0.9998 | 0.9998 | 0.9998 | 0.9998 |

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
