# Peer review of "Assessing Sustainability of the Capital and Emerging Secondary Cities of Cambodia Based on the 2018 Commune Database"

_data, 2018_

Round 1
Reviewer 1 Report
The article "Assessing Sustainability of the Capital and Emerging Secondary Cities of Cambodia Based on the 2018 Commune Database" by Puthearath Chan, submitted to Data, presents interesting results with implications for sustainability measures, not only for cities in Cambodia, but with high er scopes.
The manuscript is well-written, it is very clear in its structure and the studies' objectives are clearly presented. After refining the manuscript according to my detailed comments (minor revision), I suggest it for publication in Data. My detailed comments are both, below here as well as comments in the pdf.
detailed reviewer comments:
line 127: Africa is defined as the only region where poverty rates rise over time, with the share of
global poverty expected to reach 82% by 2030 [32].
>> from the reference, i couldn't deduce this number. could you more precisely argue that?
>> do you mean by "global" for whole Africa or world-wide? (might be misleading)
figure 4
>> please explain what the x and y axis are representing. give details for the numbers.
line 240: Source: RGC 2018
>> please give a detailed citation to the reference source
line 241-243: “By 2030, reduce adverse per capita environmental impact of
cities, including by paying special attention to air quality and municipal and other waste
management”
>> please give a reference directly for that quotation
line 243: is measuring
>> is being measured?
table 4:
>> the total weight (sum of individual weights) is 1.0006, not 1.000. please refine that or double check the individual weights (and please give the same number of decimal places for individual and total weights).
table 6:
>> also here, the total number is not equal to the sum of individual weights, 0.9994, instead of 1.000 (also here, please use the same number of decimal places and check either/or both, total and individual weights)
figure 5, figure 6, figure 7:
>> are these google-maps maps? if so, please give credit to google-maps by citing them.
line 433: The data of five cities by each indicator were sourced from the “2018 Commune Database”,
published in early 2019 by the Capital and Provincial Departments of Planning whereas the data of
PM2.5 were sourced from the “Air Quality Results in the Capital and Provinces of Cambodia”
published by the Ministry of Environment (see Appendix B).
>> please gives references.
line 478-479: The Z-scores are calculated by subtracting the mean population from each raw score and then
dividing the difference by the deviation from the population.
>> please add the variables from the Z formula (Z, x, mu, sigma) in the descriptive sentence
line 658: This The
>> Remove "This"

Author Response
Thank you very much for your appreciation, especially your time and efforts in reviewing my manuscript and providing thoughtful feedback. I revised the manuscript according to your comments and suggestions. Please see in the attached file.

Reviewer 2 Report
This paper deals with assessing Sustainability of the Capital and Emerging Secondary Cities of Cambodia. The authors base their study on the Cambodian Commune Database of 2018. The subject of the proposed paper falls within the field of green economy, urban development and management.
This paper provides a whole state of the art about the effort done over the world regarding the sustainable cities. Indeed, many references are provided. Sustainable city is then defined as a city with consideration for social, economic, environmental impact, and resilient habitat for existing populations. According to the United Nations Development Program (UNDP), being a “sustainable city” means “investment in public transport, creating green public spaces, and urban planning and management”.
This paper focuses on Cambodian sustainable city development plan which is accompanied by the green city strategic planning methodology approved in 2016, which is a step-by-step guide for municipalities, district and commune officials for transforming their cities towards green growth. This methodology holistically considers all aspects of green urban development, such as low-carbon development, climate resilience, resource efficiency, as well as social inclusion and poverty alleviation.
Hence, 18 goals, 88 targets, and 148 indicators are then identified and which were approved by the Council of Ministers in the full cabinet meeting in November 2018.
This paper gives important data about Cambodian Communes and provides interesting graphics that show how Sustainability Assessment of Cities is achieved. To assess sustainability of the capital and emerging secondary cities of Cambodia, the proposed study developed an assessment index based the United Nations and domestic concepts by focusing on the importance of the developed urban sustainability indicators in Cambodia. Therefore, the developed priority-based urban sustainability assessment index must be contained equally the number of environmental, social, and economic indicators. Hence, this study accordingly rounded up the number of three-dimensional sustainability indicators based on the priority weight.
Some statistical methods are applied to Cambodian Commune databases: the standard scores or Z-scores, other terms: Z-values, normal scores, and standard variables, is the number of standard deviations where the value of a raw score (for example, values or data observed) is higher or lower than the mean value of what is observed or measured. The above-average raw scores have positive standard scores while the below-average scores have negative standard scores. Based on these analyses, the authors give and discuss the results obtained from the comparative assessment based on the standard scores. These latter show for example that the ratio of households linked to solid waste collection services in Phnom Penh is higher than other cities, followed by Sihanoukville and Battambang. Moreover, the ratio of households installed proper toilets or adequate liquid waste storage in Battambang is much better than other four cities, followed by Poi Pet and Sihanoukville, etc.
Comments:
This article is well written and well organized.
This paper falls within the field of green economy, urban development and management. Unfortunately this article is not in the field of computer science and/or data science.
It is an interesting article that takes part in the global concern for sustainable development.
The used database contains real data. This is very interesting.
The statistical methods used in this paper and the analyses applied on the real data are very simple. I did not find a new idea or method in this paper!
Author Response
Please see it in the attachment.

Reviewer 3 Report
This paper presents a useful and practical topic. It seems ready to publish. It is well-organized and has richer contents with supportive data and results. Sustainability is one of important issues not only for developing countries but also for any countries including developed countries. My school has also high interests with this issue and has been trying to conduct several projects in the name of sustainability.
Here’s one doubt about the indicators the author instantiated and declared for analysis. Since I am a computer scientist, it is hard to judge them, but doesn’t an “education” impact the sustainability? I agree that it may not directly indicate, so I understand it is not in the list of indicators, but my thought is here: people who are educated about the sustainability issue, especially environmental issues might be aware of the necessity and will be assistant in any kinds of work. It also is able to be related to crime rate or unemployment rate, I guess. I have no data to prove it, but some cities which has good educational facilities or systems such as schools or training authorities could be important in suitability assessment. It is difficult to give you a particular formula to reveal the relations, but I’d like to hear about this thought.
There is a minor comment about content. The figure 5 has the name of cities in Korean and Cambodian only. I guess the author works with Korean research facility or uses the data provided from them. It would be better to add English names on the figure for those who have difficulties reading Korean or Cambodian.
Author Response

(The authors gave the same response as above.)

Round 2
Reviewer 2 Report
Regarding my comments of my first review, the authors give interesting responses to defend their paper. However, as I mentioned in my previous review, this article focuses on applications, descriptions, validation, and analysis of the data. But Unfortunately, as the first version, I did not find any new idea regarding data storage and access, data optimization data analysis in this new version.
The authors agree that their article was not to provide new idea or method for collecting, processing (treating), and managing the data that are mainly the aims of the Data journal. The aim of their research is to focus on the application of Commune Database in the field of sustainability. Their objective was also to show the description and reusability of the Commune Database.
It is also an application of sustainable city indicators for Cambodia developed and prioritized by Delphi and AHP by validating the sustainable city indicators with the practical data in the Commune Database.
Even if the authors use simple statistical methods, as the standard Z-scores which are commonly used, and very popular in South Korea for comparative assessment, they provide interesting results and tests. Moreover, this paper aims to share and reuse the used data in urban research fields, especially future urban research in Cambodia.
This article is an application article with interesting analysis and results. But for me, even if the article presents an application research, I should find some new issues and solutions in the conducted research.
Author Response
Please see it in the attachment.

Round 3
Reviewer 2 Report
This article is an application article with interesting analysis and results.
The authors took into account my comments and improved their paper.